# Provable Benefits of Complex Parameterizations for Structured State Space Models

Yuval Ran-Milo[I]    Eden Lumbroso[I]    Edo Cohen-Karlik[I]
Raja Giryes[I]    Amir Globerson[I II]    Nadav Cohen[I]

## Abstract

Structured state space models (SSMs), the core engine behind prominent neural networks such as S4 and Mamba, are linear dynamical systems adhering to a specified structure, most notably diagonal. In contrast to typical neural network modules, whose parameterizations are real, SSMs often use complex parameterizations. Theoretically explaining the benefits of complex parameterizations for SSMs is an open problem. The current paper takes a step towards its resolution, by establishing formal gaps between real and complex diagonal SSMs. Firstly, we prove that while a moderate dimension suffices in order for a complex SSM to express all mappings of a real SSM, a much higher dimension is needed for a real SSM to express mappings of a complex SSM. Secondly, we prove that even if the dimension of a real SSM is high enough to express a given mapping, typically, doing so requires the parameters of the real SSM to hold exponentially large values, which cannot be learned in practice. In contrast, a complex SSM can express any given mapping with moderate parameter values. Experiments corroborate our theory, and suggest a potential extension of the theory that accounts for selectivity, a new architectural feature yielding state of the art performance.[1]

## 1   Introduction

*Structured state space models* (*SSMs*) are the core engine behind prominent neural network architectures such as S4 [21], Mamba [20], LRU [41], Mega [37], S5 [50] and more [23, 31, 38, 34]. In their typical form, SSMs can be thought of as single-input single-output linear dynamical systems, wherein the state transition matrix has a specified structure, most notably diagonal [22, 23, 41, 50, 37, 20]. A salient characteristic of SSMs is that their parameterizations are often *complex* (take values in $\mathbb{C}$), in contrast to typical neural network modules whose parameterizations are conventionally real (take values in $\mathbb{R}$).

There has been mixed evidence regarding benefits of complex parameterizations over real parameterizations for SSMs. Some prior works have demonstrated that complex parameterizations are essential for strong performance [22, 41], whereas others have shown that in various settings real parameterizations lead to comparable (and in some cases better) performance [37, 20]. It was conjectured [20] that in the context of *diagonal SSMs* (namely, SSMs with diagonal state transition matrix), complex parameterizations are preferable for continuous data modalities (*e.g.*, audio, video), whereas for discrete data modalities (*e.g.*, text, DNA) real parameterizations suffice. Unfortunately, to date, formal support for this conjecture is lacking. The extent to which complex parameterizations benefit diagonal SSMs remains to be an open question.

In this paper, we take a step towards theoretically addressing the foregoing question. Specifically, we provide two theoretical contributions establishing provable benefits of complex parameterizations

---

[I]Tel Aviv University [II]Google. Correspondence to: Yuval Milo <yuvalmilo@mail.tau.ac.il>

[1]Due to lack of space, a portion of the paper is deferred to the appendices. We refer the reader to [44] for a self-contained version of the text.

38th Conference on Neural Information Processing Systems (NeurIPS 2024).

for diagonal SSMs. Our first contribution establishes that, although both real and complex diagonal SSMs are *universal*—in the sense that both can precisely express any *linear time-invariant* (*LTI*) mapping up to any time $t$ when their dimensions are equal to or greater than $t$—there is a strong separation between the SSMs in terms of expressiveness. Namely, denoting the dimensions of the real and complex SSMs by $n_\mathbb{R}$ and $n_\mathbb{C}$, respectively, we prove that for any $n_\mathbb{C}$, there are various oscillatory mappings expressible by the complex SSM which cannot be approximately expressed up to time $t$ by the real SSM unless $n_\mathbb{R}$ is on the order of $t$, which may be arbitrarily larger than $n_\mathbb{C}$. This is in stark contrast to the fact that all mappings expressible by the real SSM can be precisely expressed (up to any time) by the complex SSM whenever $n_\mathbb{C} \geq n_\mathbb{R}$.

Given the prevalence of overparameterization in machine learning, one may question how consequential the above separation (between the real and complex SSMs) is in practice. Indeed, in an overparameterized regime where a given LTI mapping is to be approximated up to a given time $t$ and $n_\mathbb{R}, n_\mathbb{C} \geq t$, universality implies that both the real and complex SSMs can precisely express the mapping up to time $t$. Accordingly, in this overparameterized regime, it is a priori unclear whether the complex SSM offers an advantage over the real SSM. Our second contribution shows that it does. Specifically, we prove a surprising result by which, if the given mapping satisfies a mild condition, then in order to approximately express the mapping up to time $t$, the real SSM must have dimension or parameter magnitude exponential in $t$. This is in stark contrast to the complex SSM, which can precisely express the given mapping up to time $t$ with dimension and parameter magnitudes that are at most linear in $t$. The aforementioned mild condition is satisfied by the canonical copy mapping, by a basic oscillatory mapping, and with high probability by a random (generic) mapping. In such important cases, practical learning of the given mapping necessitates using a complex SSM.

Our theory is corroborated by controlled experiments, demonstrating that complex parameterizations for SSMs significantly improve performance. We also evaluate SSMs with *selectivity*—a new architectural feature yielding state of the art performance [20, 31, 4, 57]. Our experiments with selectivity portray a more nuanced picture: complex parameterizations are beneficial for some tasks, whereas for others, selectivity allows real parameterizations to achieve comparable (and in some cases better) performance. These findings align with the mixed evidence reported in the literature. Moreover, they suggest a potential extension of our theory that accounts for selectivity and may elucidate this evidence, thereby fully delineating the benefits of complex parameterizations for SSMs.

## 2 Preliminaries

### 2.1 Notations

We use non-boldface lowercase letters for denoting scalars (*e.g.* $\alpha \in \mathbb{R}$, $c \in \mathbb{C}$, $n \in \mathbb{N}$), boldface lowercase letters for denoting vectors (*e.g.* $\mathbf{x} \in \mathbb{R}^n$, $\mathbf{v} \in \mathbb{C}^n$), and non-boldface uppercase letters for denoting matrices (*e.g.* $A \in \mathbb{R}^{n,m}$, $B \in \mathbb{C}^{n,m}$). Series (finite or infinite) of scalars, vectors or matrices are viewed as functions of time and denoted accordingly (*e.g.* $(\mathbf{x}(t) \in \mathbb{R}^n)_{t \in \mathbb{N}}$, $(A(t) \in \mathbb{C}^{n,m})_{t=1,2,\dots,k}$). For series of scalars, we also use as notation boldface uppercase letters (*e.g.* $\mathbf{S} = (s(t) \in \mathbb{R})_{t \in \mathbb{N}}$, $\mathbf{I} = (i(t) \in \mathbb{C})_{t=1,2,\dots,k}$). Given $k \in \mathbb{N}$ and a series of scalars $\mathbf{S}$ whose length is greater than or equal to $k$, we use $\mathbf{S}_k$ to denote the $k$th element of $\mathbf{S}$, and $\mathbf{S}_{:k}$ to denote the truncation of $\mathbf{S}$ to length $k$ (*i.e.* the series comprising the first $k$ elements of $\mathbf{S}$), allowing ourselves to regard this truncated series as a vector of dimension $k$. For $k \in \mathbb{N} \cup \{\infty\}$, we use $[k]$ as shorthand for the set $\{1, 2, \dots, k\}$. Given a complex number $c \in \mathbb{C}$, we denote its magnitude by $|c| \in \mathbb{R}_{\geq 0}$, its phase by $\arg(c) \in [0, 2\pi)$, its real part by $\Re(c) \in \mathbb{R}$, and its imaginary part by $\Im(c) \in \mathbb{R}$ (meaning $c = |c| \exp(i \arg(c)) = \Re(c) + i\Im(c)$). We let $\mathbf{0}$ and $\mathbf{1}$ stand for vectors whose entries are all zeros and all ones, respectively, with dimension to be inferred from context. The Hadamard (element-wise) product, defined between vectors or matrices of the same size, and between scalar series of the same length, is denoted by $\odot$. The convolution operator, defined between two (finite or infinite) scalar series, is denoted by $*$. Namely, given two scalar series $\mathbf{S} = (s(t))_{t \in [k]}$, $\bar{\mathbf{S}} = (\bar{s}(t))_{t \in [\bar{k}]}$ of lengths $k, \bar{k} \in \mathbb{N} \cup \{\infty\}$ respectively, $\mathbf{S} * \bar{\mathbf{S}}$ is the scalar series of length $k + \bar{k} - 1$ whose $m$th element, for $m \in [k + \bar{k} - 1]$, is given by $\sum_{t=\max\{m-\bar{k}+1,1\}}^{\min\{m,k\}} s(t)\bar{s}(m - t + 1)$.

### 2.2 Structured State Space Models

Let $\mathbb{K} = \mathbb{R}$ or $\mathbb{K} = \mathbb{C}$. A *structured state space model* (*SSM*) *of dimension* $n \in \mathbb{N}$ is parameterized by three matrices: $A \in \mathbb{K}^{n,n}$, a *state transition matrix*, which adheres to a predefined structure (*e.g.* is

constrained to be diagonal); $B \in \mathbb{K}^{n,1}$, an *input matrix*; and $C \in \mathbb{K}^{1,n}$, an *output matrix*.[23] Given the values of $A$, $B$ and $C$, the SSM realizes a mapping $\phi_{n,(A,B,C)} : \mathbb{R}^{\mathbb{N}} \to \mathbb{R}^{\mathbb{N}}$ which receives as input a real scalar series $(u(t))_{t \in \mathbb{N}}$, and produces as output a real scalar series $(y(t))_{t \in \mathbb{N}}$ defined through the following recursive formula:

$$\mathbf{x}(t) = A\mathbf{x}(t-1) + Bu(t) \ , \ \ y(t) = \Re\big(C\mathbf{x}(t)\big) \ , \ \ t \in \mathbb{N}, \tag{1}$$

where $(\mathbf{x}(t) \in \mathbb{K}^n)_{t \in \mathbb{N}}$ is a vector series of *states*, and $\mathbf{x}(0) = \mathbf{0} \in \mathbb{K}^n$. If $\mathbb{K} = \mathbb{R}$ we say that the SSM is *real*, and if $\mathbb{K} = \mathbb{C}$ we say that it is *complex*.[4] We refer to the SSM as *stable* if all eigenvalues of $A$ have magnitude strictly smaller than one; otherwise we refer to the SSM as *unstable*. For convenience, we often identify an SSM with the triplet $(A, B, C)$ holding its parameter matrices, and regard the (single column) matrices $B$ and $C^{\top}$ as vectors.

Perhaps the most prominent form of structure imposed on SSMs is *stable diagonality*, *i.e.* stability combined with diagonality [22, 23, 41, 37, 20]. Accordingly, unless stated otherwise, we assume that the state transition matrix $A$ of an SSM is diagonal and has entries with magnitude strictly smaller than one.

### 2.3 Linear Time-Invariant Mappings

Let $\phi : \mathbb{R}^{\mathbb{N}} \to \mathbb{R}^{\mathbb{N}}$ be a mapping from the space of (infinite) real scalar series to itself. We say that $\phi(\cdot)$ is *linear* if for all $\alpha \in \mathbb{R}$ and $\mathbf{S}, \bar{\mathbf{S}} \in \mathbb{R}^{\mathbb{N}}$ it holds that $\phi(\alpha \mathbf{S} + \bar{\mathbf{S}}) = \alpha \phi(\mathbf{S}) + \phi(\bar{\mathbf{S}})$. For every $k \in \mathbb{N}$, define the *$k$ step delay* $\delta_k : \mathbb{R}^{\mathbb{N}} \to \mathbb{R}^{\mathbb{N}}$ to be the operator that adds $k$ preceding zeros to the series it receives as input.[5] We say that the mapping $\phi(\cdot)$ is *linear time-invariant* (*LTI*) if it is linear, and it commutes with $\delta_k(\cdot)$ (meaning $\phi(\delta_k(\cdot)) = \delta_k(\phi(\cdot))$) for every $k \in \mathbb{N}$. It is well known [40] that if $\phi(\cdot)$ is LTI then it is given by $\phi(\mathbf{S}) = \mathbf{S} * \phi(\mathbf{I})$, where $\mathbf{I} := (1, 0, 0, \ldots) \in \mathbb{R}^{\mathbb{N}}$ is the *impulse series*, and $\phi(\mathbf{I})$ is referred to as the *impulse response* of $\phi(\cdot)$. Conversely, for any $\mathbf{R} \in \mathbb{R}^{\mathbb{N}}$, the mapping defined by $\mathbf{S} \mapsto \mathbf{S} * \mathbf{R}$ is LTI, and its impulse response is $\mathbf{R}$.

We will identify LTI mappings with their impulse responses. More specifically, for any $k \in \mathbb{N}$, we identify an LTI mapping up to time $k$, with the truncation of its impulse response to length $k$. Accordingly, for any LTI mappings $\phi(\cdot), \bar{\phi}(\cdot)$ and any $\epsilon \in \mathbb{R}_{\geq 0}$, we say that $\bar{\phi}(\cdot)$ *$\epsilon$-approximates* $\phi(\cdot)$ *up to time $k$* if $\|\phi(\mathbf{I})_{:k} - \bar{\phi}(\mathbf{I})_{:k}\|_1 \leq \epsilon$. If the latter inequality holds with $\epsilon = 0$, we also say that $\bar{\phi}(\cdot)$ *matches $\phi(\cdot)$ up to time $k$*.

Let $(A, B, C)$ be an SSM of dimension $n \in \mathbb{N}$, realizing the mapping $\phi_{n,(A,B,C)} : \mathbb{R}^{\mathbb{N}} \to \mathbb{R}^{\mathbb{N}}$ (see Section 2.2). It is straightforward to see that $\phi_{n,(A,B,C)}(\cdot)$ is LTI, and that its impulse response is given by:

$$\phi_{n,(A,B,C)}(\mathbf{I}) = \big(\Re(CB), \Re(CAB), \Re(CA^2B), \ldots \big). \tag{2}$$

For real and complex settings (*i.e.* for $\mathbb{K} = \mathbb{R}$ and $\mathbb{K} = \mathbb{C}$), we will study the extent to which varying $(A, B, C)$ as well as $n$, can lead $\phi_{n,(A,B,C)}(\cdot)$ to $\epsilon$-approximate different LTI mappings up to different times.

## 3 Theoretical Analysis

Throughout this section, we consider a real SSM $(A_{\mathbb{R}}, B_{\mathbb{R}}, C_{\mathbb{R}})$ of dimension $n_{\mathbb{R}}$ realizing the mapping $\phi_{n_{\mathbb{R}},(A_{\mathbb{R}},B_{\mathbb{R}},C_{\mathbb{R}})}(\cdot)$, and a complex SSM $(A_{\mathbb{C}}, B_{\mathbb{C}}, C_{\mathbb{C}})$ of dimension $n_{\mathbb{C}}$ realizing the mapping $\phi_{n_{\mathbb{C}},(A_{\mathbb{C}},B_{\mathbb{C}},C_{\mathbb{C}})}(\cdot)$ (see Section 2.2).

---

[2] Various SSMs include *discretization* [20, 21, 23, 16], which amounts to replacing parameter matrices by certain transformations that depend on an additional parameter $\Delta \in \mathbb{R}_{>0}$ (*e.g.*, replacing $A$ by $(I - \Delta/2 \cdot A)^{-1}(I + \Delta/2 \cdot A)$ [23, 22]). With slight modifications, our main theoretical results apply to SSMs with common discretizations—see Appendix A.1 for details.

[3] Some SSMs include an additional *feedthrough* parameter $D \in \mathbb{K}^{1,1}$ [41]. With feedthrough, the expression for $y(t)$ in Equation (1) becomes $\Re(C\mathbf{x}(t) + Du(t))$. Our theory essentially applies as is to SSMs with feedthrough—see Appendix A.2 for details.

[4] It is possible to consider a hybrid setting where $A$ is allowed to be complex while $B$ and $C$ are restricted to be real. This setting enjoys all provable benefits of the complex setting ($\mathbb{K} = \mathbb{C}$)—see Appendix A.3 for details.

[5] That is, for any $\mathbf{S} = (s(t))_{t \in \mathbb{N}} \in \mathbb{R}^{\mathbb{N}}$, $\delta_k(\mathbf{S}) \in \mathbb{R}^{\mathbb{N}}$ is the series whose $m$th element, for $m \in \mathbb{N}$, equals $s(m - k)$ if $m > k$ and 0 otherwise.

## 3.1 Universality

It is known (see, *e.g.*, [14]) that the real SSM is *universal*, in the sense that it can precisely express any LTI mapping up to any time $t$ when its dimension is equal to or greater than $t$. Trivially, this implies the same for the complex SSM. Proposition 1 below formalizes these facts for completeness.

**Proposition 1.** *Let $\phi : \mathbb{R}^{\mathbb{N}} \to \mathbb{R}^{\mathbb{N}}$ be an arbitrary LTI mapping, and let $t \in \mathbb{N}$. Then, the following holds for both $\mathbb{K} = \mathbb{R}$ and $\mathbb{K} = \mathbb{C}$. If $n_{\mathbb{K}} \geq t$, there exist assignments for $(A_{\mathbb{K}}, B_{\mathbb{K}}, C_{\mathbb{K}})$ with which $\phi_{n_{\mathbb{K}}, (A_{\mathbb{K}}, B_{\mathbb{K}}, C_{\mathbb{K}})}(\cdot)$ matches $\phi(\cdot)$ up to time $t$.*[6]

*Proof sketch (proof in Appendix D.1).* Beginning with the real SSM ($\mathbb{K} = \mathbb{R}$), the proof shows that $\phi_{n_{\mathbb{R}}, (A_{\mathbb{R}}, B_{\mathbb{R}}, C_{\mathbb{R}})}(\cdot)$ matches $\phi(\cdot)$ up to time $t$ if $V(A_{\mathbb{R}})(C_{\mathbb{R}}^{\top} \odot B_{\mathbb{R}}) = \phi(\mathbf{I})_{:t}$, where $V(A_{\mathbb{R}})$ is a Vandermonde matrix that has full rank when the diagonal entries of $A_{\mathbb{R}}$ are distinct. Assigning $A_{\mathbb{R}}$ this way, there must exist $\mathbf{v} \in \mathbb{R}^{n_{\mathbb{R}}}$ with which $V(A_{\mathbb{R}})\mathbf{v} = \phi(\mathbf{I})_{:t}$. Assigning $B_{\mathbb{R}} = \mathbf{v}$ and $C_{\mathbb{R}}^{\top} = \mathbf{1}$ concludes the proof for the real SSM. The complex SSM ($\mathbb{K} = \mathbb{C}$) can be treated analogously. $\square$

## 3.2 Separation in Expressiveness

Proposition 2 and Theorem 1 below together establish that although both the real and complex SSMs are universal (see Section 3.1), there is a strong separation between the two in terms of expressiveness. Proposition 2 formalizes an obvious fact: all mappings expressible by the real SSM can be precisely expressed (up to any time) by the complex SSM whenever $n_{\mathbb{C}} \geq n_{\mathbb{R}}$ (*i.e.*, whenever the dimension of the complex SSM is equal to or greater than the dimension of the real SSM). Theorem 1 proves a much less obvious result: for any $n_{\mathbb{C}}$, there are various oscillatory mappings (*i.e.* mappings with oscillatory impulse responses) expressible by the complex SSM which cannot be approximately expressed up to time $t$ by the real SSM unless $n_{\mathbb{R}}$ is on the order of $t$, which may be arbitrarily larger than $n_{\mathbb{C}}$.

**Proposition 2.** *Consider an arbitrary assignment for $(A_{\mathbb{R}}, B_{\mathbb{R}}, C_{\mathbb{R}})$, and assume that $n_{\mathbb{C}} \geq n_{\mathbb{R}}$. Then, there exist assignments for $(A_{\mathbb{C}}, B_{\mathbb{C}}, C_{\mathbb{C}})$ with which $\phi_{n_{\mathbb{C}}, (A_{\mathbb{C}}, B_{\mathbb{C}}, C_{\mathbb{C}})}(\cdot) = \phi_{n_{\mathbb{R}}, (A_{\mathbb{R}}, B_{\mathbb{R}}, C_{\mathbb{R}})}(\cdot)$.*

*Proof.* It suffices to prove the sought after result for $n_{\mathbb{C}} = n_{\mathbb{R}}$, since one can effectively reduce the dimension of the complex SSM by zeroing out entries of $B_{\mathbb{C}}$ (or $C_{\mathbb{C}}$). Assuming that $n_{\mathbb{C}} = n_{\mathbb{R}}$, we may assign to $(A_{\mathbb{C}}, B_{\mathbb{C}}, C_{\mathbb{C}})$ the values of $(A_{\mathbb{R}}, B_{\mathbb{R}}, C_{\mathbb{R}})$. Under this assignment $\phi_{n_{\mathbb{C}}, (A_{\mathbb{C}}, B_{\mathbb{C}}, C_{\mathbb{C}})}(\cdot) = \phi_{n_{\mathbb{R}}, (A_{\mathbb{R}}, B_{\mathbb{R}}, C_{\mathbb{R}})}(\cdot)$, as required. $\square$

**Theorem 1.** *Let $t \in \mathbb{N}$ and $\epsilon \in \mathbb{R}_{\geq 0}$. Assume without loss of generality that $n_{\mathbb{C}} = 1$,[7] in which case $A_{\mathbb{C}}$, $B_{\mathbb{C}}$ and $C_{\mathbb{C}}$ can be regarded as scalars. Suppose $|\sin(\arg(A_{\mathbb{C}}))| \geq 0.2$, $|A_{\mathbb{C}}| \geq 0.5^{1/t}$ and $|B_{\mathbb{C}} \cdot C_{\mathbb{C}}| \geq 1$. Then, if $\phi_{n_{\mathbb{R}}, (A_{\mathbb{R}}, B_{\mathbb{R}}, C_{\mathbb{R}})}(\cdot)$ $\epsilon$-approximates $\phi_{n_{\mathbb{C}}, (A_{\mathbb{C}}, B_{\mathbb{C}}, C_{\mathbb{C}})}(\cdot)$ up to time $t$, it must be that $n_{\mathbb{R}} \geq \lfloor t/9 \rfloor - 1 - 4\epsilon$.*

*Proof sketch (proof in Appendix D.2).* The idea behind the proof is as follows. The complex SSM realizes an oscillatory mapping, in the sense that elements $1, 3, \ldots, 2\lceil t/2 \rceil - 1$ of its impulse response alternate $\Theta(t)$ times between being greater than or equal to $1/4$, and being smaller than or equal to $-1/4$. The real SSM on the other hand is limited in its ability to realize oscillations, insofar as elements $1, 3, \ldots, 2\lceil t/2 \rceil - 1$ of the impulse response it gives rise to are a linear combination of decaying exponentials, and therefore can change sign at most $\mathcal{O}(n_{\mathbb{R}})$ times. Combining these two observations leads to the desired result. $\square$

## 3.3 Separation in Practical Learnability

Let $\phi : \mathbb{R}^{\mathbb{N}} \to \mathbb{R}^{\mathbb{N}}$ be an LTI mapping with bounded impulse response, which we would like to $\epsilon$-approximate up to time $t$ for some $\epsilon \in \mathbb{R}_{\geq 0}$ and $t \in \mathbb{N}$. Assume that the dimensions of the real and complex SSMs are greater than or equal to $t$. By Proposition 1, both the real and complex SSMs can express mappings that match $\phi(\cdot)$ up to time $t$, and in particular that achieve the desired approximation. The current subsection establishes that despite this parity in terms of expressiveness, there is a strong separation between the real and complex SSMs in terms of *practical learnability*.

---

[6]It is possible to strengthen this result, *e.g.*, by showing that the requirement $n_{\mathbb{K}} \geq t$ can be replaced by $n_{\mathbb{K}} \geq \lceil (t+1)/2 \rceil$ when $\mathbb{K} = \mathbb{C}$. We omit details, as the current form of the result suffices for our purposes.

[7]This does not limit generality since one can effectively reduce the dimension of the complex SSM by zeroing out entries of $B_{\mathbb{C}}$ (or $C_{\mathbb{C}}$).

Section 3.3.1 proves that under a mild condition on $\phi(\cdot)$, in order for the real SSM to achieve the desired approximation, either its dimension or the magnitude of its parameters must be exponential in $t$. Section 3.3.2 then explains that such exponentiality impedes practical learning via gradient descent. Finally, Section 3.3.3 proves that in stark contrast to the real SSM, the complex SSM can achieve the desired approximation with dimension and parameter magnitudes that are at most linear in $t$.

### 3.3.1 Real Parameterizations Suffer from Exponentiality

Definition 1 below formalizes the notion of *forward difference* for a real scalar series—a discrete analogue of derivative for a differentiable real function. Our main theoretical result, Theorem 2, then establishes that if forward differences associated with $\phi(\mathbf{I})$—the impulse response of $\phi(\cdot)$—satisfy a certain condition, then in order for the real SSM to express a mapping that $\epsilon$-approximates $\phi(\cdot)$ up to time $t$, either the dimension of the real SSM $n_{\mathbb{R}}$ or the magnitude of its parameters $(B_{\mathbb{R}}, C_{\mathbb{R}})$ must be exponential in $t$. Roughly speaking, the aforementioned condition on forward differences associated with $\phi(\mathbf{I})$ is that there exists some $d \in \Theta(t)$ such that the $d$th forward difference of the restriction of $\phi(\mathbf{I})$ to either odd or even elements has magnitude greater than $2^d \epsilon$. Perhaps surprisingly, this condition is especially mild, as the magnitude of the $d$th forward difference of a real scalar series typically scales exponentially with $d$. Several important cases where the condition is satisfied are presented below.

**Definition 1.** Let $\mathbf{S}$ be a real scalar series of length $k \in \mathbb{N} \cup \{\infty\}$. The *forward difference* of $\mathbf{S}$, denoted $\mathbf{S}^{(1)}$, is the scalar series of length $k - 1$ whose $m$th element, for $m \in [k - 1]$, is given by $\mathbf{S}_{m+1} - \mathbf{S}_m$. For $d \in \{2, 3, \ldots, k - 1\}$, the *$d$th forward difference* of $\mathbf{S}$, denoted $\mathbf{S}^{(d)}$, is recursively defined to be the forward difference of $\mathbf{S}^{(d-1)}$.

**Theorem 2.** *Suppose $\phi_{n_{\mathbb{R}},(A_{\mathbb{R}},B_{\mathbb{R}},C_{\mathbb{R}})}(\cdot)$ $\epsilon$-approximates $\phi(\cdot)$ up to time $t$. Then:*

$$n_{\mathbb{R}} \| C_{\mathbb{R}}^\top \odot B_{\mathbb{R}} \|_\infty \geq \max_{\substack{d,m \in \mathbb{N},\, d+m \leq \lfloor t/2 \rfloor \\ \sigma \in \{odd,\, even\}}} \left\{ 2^{d+2\min\{d,m\}} \left( 2^{-d} \big| (\phi(\mathbf{I})|_\sigma)_m^{(d)} \big| - \epsilon \right) \right\}, \qquad (3)$$

*where: $\phi(\mathbf{I})|_{odd}$ and $\phi(\mathbf{I})|_{even}$ are the restrictions of the impulse response $\phi(\mathbf{I})$ to odd and even elements, respectively; and $(\phi(\mathbf{I})|_{odd})_m^{(d)}$ and $(\phi(\mathbf{I})|_{even})_m^{(d)}$ stand for the $m$th element of the $d$th forward difference of $\phi(\mathbf{I})|_{odd}$ and $\phi(\mathbf{I})|_{even}$, respectively.*

*Proof sketch (proof in Appendix D.3).* The idea behind the proof is as follows. The restrictions of the impulse response of $\phi_{n_{\mathbb{R}},(A_{\mathbb{R}},B_{\mathbb{R}},C_{\mathbb{R}})}(\cdot)$ to odd and even elements—*i.e.*, $\phi_{n_{\mathbb{R}},(A_{\mathbb{R}},B_{\mathbb{R}},C_{\mathbb{R}})}(\mathbf{I})|_{odd}$ and $\phi_{n_{\mathbb{R}},(A_{\mathbb{R}},B_{\mathbb{R}},C_{\mathbb{R}})}(\mathbf{I})|_{even}$, respectively—are each a linear combination of $n_{\mathbb{R}}$ decaying exponentials, where the coefficients of the linear combination have absolute value no greater than $\| C_{\mathbb{R}}^\top \odot B_{\mathbb{R}} \|_\infty$. Forward differences of decaying exponentials are exponentially small. Therefore, by linearity of forward differences, requiring $\phi_{n_{\mathbb{R}},(A_{\mathbb{R}},B_{\mathbb{R}},C_{\mathbb{R}})}(\mathbf{I})|_{odd}$ or $\phi_{n_{\mathbb{R}},(A_{\mathbb{R}},B_{\mathbb{R}},C_{\mathbb{R}})}(\mathbf{I})|_{even}$ to have a forward difference that is not exponentially small implies an exponentially large lower bound on $n_{\mathbb{R}} \| C_{\mathbb{R}}^\top \odot B_{\mathbb{R}} \|_\infty$. When $\phi_{n_{\mathbb{R}},(A_{\mathbb{R}},B_{\mathbb{R}},C_{\mathbb{R}})}(\cdot)$ $\epsilon$-approximates $\phi(\cdot)$ up to time $t$, forward differences of $\phi_{n_{\mathbb{R}},(A_{\mathbb{R}},B_{\mathbb{R}},C_{\mathbb{R}})}(\mathbf{I})|_{odd}$ and $\phi_{n_{\mathbb{R}},(A_{\mathbb{R}},B_{\mathbb{R}},C_{\mathbb{R}})}(\mathbf{I})|_{even}$ are close to those of $\phi(\mathbf{I})|_{odd}$ and $\phi(\mathbf{I})|_{even}$, respectively. We thus conclude that if $\phi(\mathbf{I})|_{odd}$ or $\phi(\mathbf{I})|_{even}$ has a forward difference that is not especially small, then $n_{\mathbb{R}} \| C_{\mathbb{R}}^\top \odot B_{\mathbb{R}} \|_\infty$ must be exponentially large. This conclusion is formalized via Equation (3), the sought after result. $\qquad \square$

**Special cases.** Theorem 2 implies that the real SSM suffers from exponentiality (namely, its dimension $n_{\mathbb{R}}$ or the magnitude of its parameters $(B_{\mathbb{R}}, C_{\mathbb{R}})$ must be exponential in $t$ in order for it to express a mapping that $\epsilon$-approximates $\phi(\cdot)$ up to time $t$) in various important cases. Indeed, Corollaries 1 and 2 below respectively show that the real SSM suffers from exponentiality when $\phi(\cdot)$ is a canonical copy (delay) mapping, and with high probability when $\phi(\cdot)$ is a random (generic) mapping. In light of Theorem 1 (namely, of the inability of the real SSM to compactly approximate various oscillatory mappings expressible by the complex SSM), it is natural to ask if the real SSM suffers from exponentiality in cases where $\phi(\cdot)$ is oscillatory, *i.e.* where its impulse response oscillates. Corollary 3 below shows that exponentiality indeed transpires in a case where $\phi(\cdot)$ is a basic oscillatory mapping. On the other hand, there are simple cases where $\phi(\cdot)$ is oscillatory yet exponentiality does not take

place, *e.g.* the case where the impulse response of $\phi(\cdot)$ is $(+1, -1, +1, -1, \ldots)$.[8] Precise delineation of the type of oscillations that lead to exponentiality is deferred to future work (see Section 6).

**Corollary 1.** *Suppose $t \geq 9$ and $\phi(\cdot) = \delta_{\lfloor (t-1)/2 \rfloor}(\cdot)$, where as defined in Section 2.3, $\delta_{\lfloor (t-1)/2 \rfloor}(\cdot)$ is the $\lfloor (t-1)/2 \rfloor$ step delay mapping. Assume also that $\epsilon \leq 1/(8\sqrt{t})$. Then, if $\phi_{n_\mathbb{R}, (A_\mathbb{R}, B_\mathbb{R}, C_\mathbb{R})}(\cdot)$ $\epsilon$-approximates $\phi(\cdot)$ up to time $t$, it must hold that:*

$$n_\mathbb{R} \| C_\mathbb{R}^\top \odot B_\mathbb{R} \|_\infty \geq 2^{t/2}/(32\sqrt{t}). \tag{4}$$

*Proof sketch (proof in Appendix D.4).* The proof computes forward differences associated with $\delta_{\lfloor (t-1)/2 \rfloor}(\mathbf{I})$ (impulse response of $\delta_{\lfloor (t-1)/2 \rfloor}(\cdot)$), and plugs them into Theorem 2 (Equation (3)). $\square$

**Corollary 2.** *Let $\alpha \in \mathbb{R}_{>0}$, and let $\mathbf{R} \in \mathbb{R}^\mathbb{N}$ be generated by a random process where each element of $\mathbf{R}$ is independently drawn from a uniform distribution over the interval $[-\alpha, \alpha]$. Suppose that $t \geq 8$ and that $\phi(\cdot)$ is the mapping whose impulse response is $\mathbf{R}$ (i.e., $\phi(\cdot)$ is defined by $\phi(\mathbf{S}) = \mathbf{S} * \mathbf{R}$). Let $p \in (0, 1)$, and assume that $\epsilon \leq \alpha \sqrt{p/t}$. Then, with probability at least $1 - p$, if $\phi_{n_\mathbb{R}, (A_\mathbb{R}, B_\mathbb{R}, C_\mathbb{R})}(\cdot)$ $\epsilon$-approximates $\phi(\cdot)$ up to time $t$, it must hold that:*

$$n_\mathbb{R} \| C_\mathbb{R}^\top \odot B_\mathbb{R} \|_\infty \geq 2^{t/2} \alpha \sqrt{p}/(8\sqrt{t}). \tag{5}$$

*Proof sketch (proof in Appendix D.5).* The proof derives lower bounds (holding with probability at least $1-p$) on forward differences associated with $\mathbf{R}$, and plugs them into Theorem 2 (Equation (3)). $\square$

**Corollary 3.** *Suppose that $\phi(\mathbf{I}) = (+1, \, 0, \, -1, \, 0, \, +1, \, 0, \, -1, \, 0, \ldots)$ and $\epsilon \leq 0.5$. Then, if $\phi_{n_\mathbb{R}, (A_\mathbb{R}, B_\mathbb{R}, C_\mathbb{R})}(\cdot)$ $\epsilon$-approximates $\phi(\cdot)$ up to time $t$, it must hold that:*

$$n_\mathbb{R} \| C_\mathbb{R}^\top \odot B_\mathbb{R} \|_\infty \geq 2^{3t/4 - 4}. \tag{6}$$

*Proof sketch (proof in Appendix D.6).* The proof computes forward differences associated with the series $(+1, \, 0, \, -1, \, 0, \, +1, \, 0, \, -1, \, 0, \ldots)$, and plugs them into Theorem 2 (Equation (3)). $\square$

### 3.3.2 Exponentiality Impedes Practical Learning

For any value of $t$ that is not especially small, exponentiality in $t$ for the real SSM as put forth in Section 3.3.1—*i.e.*, exponentiality in $t$ of the dimension of the real SSM $n_\mathbb{R}$ or the magnitude of its parameters $(B_\mathbb{R}, C_\mathbb{R})$—impedes practical learning. This impediment is obvious in the case where $n_\mathbb{R}$ is exponential in $t$ (in this case, it is impractical to even store the parameters of the real SSM, let alone learn them). Appendix B treats the complementary case, *i.e.* it shows that learning is impractical when the required values for the parameters $(B_\mathbb{R}, C_\mathbb{R})$ are exponential in $t$ (this is deferred to an appendix due to space constraints). The results of Section 3.3.1 therefore imply that the real SSM cannot practically learn a mapping that $\epsilon$-approximates $\phi(\cdot)$ up to time $t$ under important choices of $\phi(\cdot)$.

### 3.3.3 Complex Parameterizations Do Not Suffer from Exponentiality

Section 3.3.1 established that under a mild condition on $\phi(\cdot)$, in order for the real SSM to express a mapping that $\epsilon$-approximates $\phi(\cdot)$ up to time $t$, either the dimension of the real SSM $n_\mathbb{R}$ or the magnitude of its parameters $(B_\mathbb{R}, C_\mathbb{R})$ must be exponential in $t$. Proposition 3 below proves that in stark contrast, for any choice of $\phi(\cdot)$ (whose impulse response $\phi(\mathbf{I})$ is bounded), the complex SSM can express mappings that match $\phi(\cdot)$ up to time $t$ with dimension $n_\mathbb{C}$ and magnitude of parameters $(B_\mathbb{C}, C_\mathbb{C})$ that are at most linear in $t$.

**Proposition 3.** *For any choice of $n_\mathbb{C}$ greater than or equal to $t$, there exist assignments for $(A_\mathbb{C}, B_\mathbb{C}, C_\mathbb{C})$ with which $\phi_{n_\mathbb{C}, (A_\mathbb{C}, B_\mathbb{C}, C_\mathbb{C})}(\cdot)$ matches $\phi(\cdot)$ up to time $t$, and wherein: (i) $\|B_\mathbb{C}\|_2 \leq 2\|\phi(\mathbf{I})_{:t}\|_2$; and (ii) $\|C_\mathbb{C}^\top\|_2 \leq 1$.[9]*

---

[8]To see that exponentiality does not take place when $\phi(\mathbf{I}) = (+1, -1, +1, -1, \ldots)$, note that in this case, with any $n_\mathbb{R}$, $\phi_{n_\mathbb{R}, (A_\mathbb{R}, B_\mathbb{R}, C_\mathbb{R})}(\cdot)$ $\epsilon$-approximates $\phi(\cdot)$ up to time $t$ whenever: $C_\mathbb{R}^\top \odot B_\mathbb{R}$ holds one in its first entry and zeros elsewhere; and $A_\mathbb{R}$ holds $\min\{0, (-1 + \epsilon/t^2)\}$ in its first diagonal entry.

[9]It is possible to develop a variant of this result that only requires $n_\mathbb{C} \geq \lceil (t+1)/2 \rceil$. We omit details, as the current form of the result suffices for our purposes.

*Proof sketch (proof in Appendix D.7).* The proof employs the theory of *discrete Fourier transform (DFT)*. It begins by assigning (scaled versions of) the $t$th roots of unity to the diagonal entries of $A_{\mathbb{C}}$. Then, it uses the inverse DFT formula to derive assignments for $B_{\mathbb{C}}$ and $C_{\mathbb{C}}$ leading $\phi_{n_{\mathbb{C}},(A_{\mathbb{C}},B_{\mathbb{C}},C_{\mathbb{C}})}(\cdot)$ to match $\phi(\cdot)$ up to time $t$. Finally, the proof applies Plancheral theorem to show that the derived assignments for $B_{\mathbb{C}}$ and $C_{\mathbb{C}}$ satisfy the desired criteria. $\square$

## 4    Experiments

This section presents controlled experiments corroborating our theory. Section 4.1 demonstrates that complex parameterizations significantly improve performance of SSMs in the theoretically analyzed setting. Section 4.2 shows that this improvement extends to a real-world setting beyond our theory. Finally, Section 4.3 evaluates SSMs with *selectivity*—a new architectural feature yielding state of the art performance [20, 31, 4, 57]. The experiments with selectivity portray a nuanced picture: complex parameterizations are beneficial for some tasks, whereas for others, selectivity allows real parameterizations to achieve comparable (and in some cases better) performance. These findings align with mixed evidence reported in the literature (see Section 1). Moreover, they suggest a potential extension of our theory that accounts for selectivity and may elucidate this evidence, thereby fully delineating the benefits of complex parameterizations for SSMs.

For conciseness, we defer some of the details behind our implementation to Appendix F. Code for reproducing our experiments is available at `https://github.com/edenlum/SSMComplexParamBenefits`.

### 4.1    Theoretically Analyzed Setting

To empirically demonstrate our theoretical findings, we trained the analyzed real and complex SSMs (see Section 2.2) to approximate up to time $t$ the mapping $\phi(\cdot)$ (see Section 2.3), with $t$ varying and with the following choices for $\phi(\cdot)$: the canonical copy (delay) mapping from Corollary 1; the random (generic) mapping from Corollary 2; and the basic oscillatory mapping from Corollary 3. Throughout, the dimension of the real or complex SSM ($n_{\mathbb{R}}$ or $n_{\mathbb{C}}$, respectively) was set to at least $t$, which, by universality (Section 3.1), implies that the SSM can express a mapping that precisely matches $\phi(\cdot)$ up to time $t$. Our theory (Section 3.3) establishes that despite this parity between the real and complex SSMs in terms of expressiveness, there is a strong separation between the SSMs in terms of practical learnability. In particular, with the above choices of $\phi(\cdot)$, there are exponential barriers that apply only to the real SSM, and prevent its training from being able to yield a mapping that closely approximates $\phi(\cdot)$ up to time $t$. Tables 1 and 2 present results obtained with the real and complex SSMs, respectively. They confirm the predictions of our theory.

Table 1: In accordance with our theory, the analyzed real SSM (see Section 2.2) cannot practically learn to closely approximate $\phi(\cdot)$ up to time $t$ under important choices of $\phi(\cdot)$, even when $t$ is moderate. This table reports the approximation error attained by the real SSM, *i.e.* the minimum $\epsilon$ with which a mapping learned by the real SSM $\epsilon$-approximates $\phi(\cdot)$ up to time $t$ (see Section 2.3), when $t = 32$ and $\phi(\cdot)$ varies over the following possibilities: the canonical copy (delay) mapping from Corollary 1; the random (generic) mapping from Corollary 2; and the basic oscillatory mapping from Corollary 3. Learning was implemented by applying one of three possible gradient-based optimizers—Adam [29], AdamW [36] or RAdam [33]—to a loss as in our theory (see Appendix C). For each choice of $\phi(\cdot)$, reported approximation errors are normalized (scaled) such that a value of one is attained by the trivial zero mapping. Each configuration was evaluated with five random seeds, and its reported approximation error is the minimum (best) that was attained. The dimension of the real SSM ($n_{\mathbb{R}}$) was set to $1024$; other choices of dimension led to qualitatively identical results. For further implementation details see Appendix F.1.

| Optimizer | Approx. of Copy | Approx. of Random | Approx. of Oscillatory |
|-----------|-----------------|-------------------|------------------------|
| Adam      | 0.767           | 0.536             | 0.812                  |
| AdamW     | 0.772           | 0.541             | 0.809                  |
| RAdam     | 0.767           | 0.538             | 0.811                  |

### 4.2    Real-World Setting

To empirically demonstrate the benefits of complex parameterizations for SSMs in settings beyond our theory, we evaluated the prominent S4 neural network architecture [21] on the real-world sequential

Table 2: In contrast to the analyzed real SSM, and in alignment with our theory, the analyzed complex SSM (see Section 2.2) can practically learn to closely approximate $\phi(\cdot)$ up to time $t$ under important choices of $\phi(\cdot)$ and various choices of $t$. This table reports approximation errors attained by the complex SSM. It adheres to the description of Table 1, with the following exceptions (all designed to stress the superiority of the complex SSM over the real SSM): *(i)* only Adam optimizer was used; *(ii)* in addition to 32, $t$ also took the values 64, 128 and 256; *(iii)* for each configuration, the reported approximation error is the maximum (worst) that was achieved across the random seeds; and *(iv)* the dimension of the complex SSM ($n_{\mathbb{C}}$) was set to $t$ (higher dimensions led to qualitatively identical results). For further implementation details see Appendix F.1.

| $t$ | **Approx. of Copy** | **Approx. of Random** | **Approx. of Oscillatory** |
|-----|------------------|---------------------|--------------------------|
| 32  | $1.6 \times 10^{-5}$ | $6.3 \times 10^{-5}$ | $1.6 \times 10^{-4}$ |
| 64  | $1.7 \times 10^{-5}$ | $1.6 \times 10^{-5}$ | $3.7 \times 10^{-4}$ |
| 128 | $4.9 \times 10^{-5}$ | $4.8 \times 10^{-5}$ | $2.6 \times 10^{-3}$ |
| 256 | $6.8 \times 10^{-4}$ | $7.6 \times 10^{-4}$ | $1.7 \times 10^{-3}$ |

CIFAR-10 dataset from the widely recognized Long Range Arena benchmark [52]. Our implementation is based on the official S4 repository,[10] where unless stated otherwise, hyperparameters (pertaining to the neural network architecture and its training) were kept at their default values. A single run with complex parameterization yielded a test accuracy of $89.10\%$, significantly higher than the highest test accuracy of $78.27\%$ attained with real parameterization across three random seeds. Modifying the optimizer and initialization scheme with the real parameterization did not improve the test accuracy—see Appendix F.2 for details. The overarching conclusion from our theory—namely, that SSMs benefit from complex parameterizations—thus extends to this real-world setting.

### 4.3 Selectivity

A new architectural feature for SSMs that yields state of the art performance is *selectivity* [20, 31, 4, 57]. In its original form—proposed as part of the Mamba neural network architecture [20]— selectivity amounts to replacing the parameters $B$ and $C$ (see Section 2.2), as well as an additional *discretization* parameter $\Delta \in \mathbb{R}_{>0}$,[2] by certain functions of the input $(u(t))_{t \in \mathbb{N}}$. We empirically study the impact of complex parameterizations on SSMs with selectivity by evaluating a Mamba neural network on two synthetic tasks regarded as canonical in the SSM literature [27, 20]: *(i) copy*, which was shown by our theory (Section 3.3) and earlier experiments (Section 4.1) to pose difficulties for real parameterizations in SSMs with no selectivity; and *(ii) induction-head*, which can be seen as a generalization of copy in which the delay is input-specified (rather than being fixed). Our implementation is based on a widely adopted Mamba repository easily amenable to modification.[11] Unless stated otherwise, repository hyperparameters (pertaining to the neural network architecture and its training) were kept at their default values. Further details concerning our implementation, including detailed descriptions of the copy and induction-head tasks, can be found in Appendix F.3.

Our first experiment with the Mamba neural network compared real and complex parameterizations for the underlying SSMs, with selectivity included. On the copy task, across three random seeds for each configuration, the highest accuracy attained with the real parameterization was $80.17\%$, whereas the lowest accuracy attained with the complex parameterization was $93.05\%$. This gap in performance in favor of the complex parameterization aligns with our theoretical and empirical findings for SSMs without selectivity. In stark contrast, on the induction-head task, there is no such gap (in fact, there is a slight advantage to the real parameterization): across three random seeds for each configuration, accuracies attained with the real parameterization ranged between $97.35\%$ and $98.3\%$, whereas those attained with the complex parameterization ranged between $93.93\%$ and $97.64\%$. These results align with mixed evidence reported in the SSM literature, by which complex parameterizations are essential for strong performance on some tasks [22, 41, 22], whereas on others, real parameterizations lead to comparable (and in some cases better) performance [37, 20].

To gain insight into the induction-head task not benefiting from the complex parameterization, we conducted an ablation experiment with partial versions of selectivity (*i.e.*, where not all of the parameters $B$, $C$ and $\Delta$ were replaced by functions of the input). The results of this experiment, reported in Table 3, reveal that when selectivity is fully or partially removed (more precisely, when

---

[10]https://github.com/state-spaces/s4 (Apache-2.0 license).

[11] https://github.com/alxndrTL/mamba.py (MIT license).

Table 3: Ablation experiment demonstrating that real parameterizations can compare (favorably) to complex parameterizations for SSMs with selectivity, but complex parameterizations become superior when selectivity is fully or partially removed. This table reports test accuracies attained by a Mamba neural network [20] on a synthetic induction-head task regarded as canonical in the SSM literature [27, 20]. Evaluation included multiple configurations for the SSMs underlying the neural network. Each configuration corresponds to either real or complex parameterization, and to a specific partial version of selectivity—*i.e.*, to a specific combination of parameters that are selective (replaced by functions of the input), where the parameters that may be selective are: the input matrix $B$; the output matrix $C$; and a discretization parameter $\Delta$. For each configuration, the highest and lowest accuracies attained across three random seeds are reported. Notice that when both $B$ and $C$ are selective, the real parameterization compares (favorably) to the complex parameterization, whereas otherwise, the complex parameterization is superior. For further implementation details see Appendix F.3.

| Selective $B$ | Selective $C$ | Selective $\Delta$ | Real Accuracy (%) | Complex Accuracy (%) |
|---|---|---|---|---|
| Yes | Yes | Yes | 97.35 to 98.3 | 93.93 to 97.64 |
| Yes | Yes | No | 97.9 to 98.62 | 90.18 to 95.86 |
| Yes | No | Yes | 61.82 to 71.28 | 91.93 to 96.77 |
| Yes | No | No | 49.91 to 52.5 | 58.93 to 73.77 |
| No | Yes | Yes | 56.78 to 69.54 | 92.52 to 96.91 |
| No | Yes | No | 41.01 to 43.89 | 57.44 to 64.67 |
| No | No | Yes | 15.48 to 26.33 | 68.54 to 79.71 |
| No | No | No | 23.86 to 29.62 | 37.61 to 50.19 |

$B$, $C$ or both are not replaced by functions of the input), the complex parameterization regains its advantage. This suggests that selectivity, which is not covered by our theory, may be the key factor enabling real parameterizations to perform as well as complex parameterizations for SSMs on certain tasks. In other words, selectivity may be the dominant factor behind the aforementioned evidence in the SSM literature being mixed. In Section 7 we outline a potential extension of our theory that accounts for selectivity and may elucidate this evidence, thereby fully delineating the benefits of complex parameterizations for SSMs.

## 5  Related Work

SSMs are closely related to linear dynamical systems—a classic object of study in areas such as systems theory [3] and control theory [51]. Although there exists extensive literature concerning properties of real and complex linear dynamical systems [10, 5, 6, 9], this literature does not readily establish benefits of complex parameterizations for SSMs, primarily due to the following reasons: *(i)* the output of a complex SSM is turned real by disregarding imaginary components (see Section 2.2), therefore it differs from a complex linear dynamical system; and *(ii)* the structures typically imposed on state transition matrices of SSMs (*e.g.* stable diagonality; see Section 2.2) are generally uncommon in the literature on linear dynamical systems.

SSMs can be viewed as a special case of recurrent neural networks [48], which received significant theoretical attention [46, 39, 25, 11, 53]. In this context, several works focused specifically on SSMs [42, 30, 24, 2, 28, 58, 12, 32, 55]. However, to our knowledge, the only prior work to formally and explicitly treat benefits of complex parameterizations for SSMs is [42]. The treatment of [42] (see Section 4.1 therein) can be viewed as a special case of ours. Indeed, [42] considered a task that, using our notation (see Section 2.2), amounts to linearly reconstructing an input element $u(t)$ from the state $\mathbf{x}(t')$ of an SSM, where $t, t' \in \mathbb{N}, t' \geq t$. This is equivalent to assigning the output matrix of the SSM $C$ such that the mapping realized by the SSM $\phi_{n,(A,B,C)}(\cdot)$ is a canonical copy (delay) mapping. Roughly speaking, [42] showed that this task requires linear operations with exponential parameters if the SSM is real, whereas linear operations with moderate parameters suffice if the SSM is complex. The same result follows from our Corollary 1 and Proposition 3. We stress that our theory goes far beyond this result, for example in that it covers various mappings beyond copy, including a random (generic) mapping—see Section 3.3 for details.

With regards to related empirical work, the literature includes several experimental comparisons between real and complex parameterizations for SSMs [20, 22, 42]. Nonetheless, to our knowledge, the controlled experiments we conducted (see Section 4) are reported herein for the first time.

# 6 Limitations

While this paper offers meaningful contributions regarding benefits of complex parameterizations for SSMs, it is important to acknowledge several of its limitations. First, while we establish a separation between real and complex parameterizations in terms of expressiveness (Section 3.2), we do not quantify how prevalent this separation is, *i.e.*, what proportion of the mappings expressible with complex parameterizations cannot be compactly approximated with real parameterizations. Second, while we prove that real parameterizations suffer from exponentiality that impedes practical learning (Sections 3.3.1 and 3.3.2), and that complex parameterizations do not suffer from exponentiality (Section 3.3.3), we do not formally establish practical learnability with complex parameterizations—our evidence for this is purely empirical (Section 4). Third, our theory does not treat all fundamental aspects of learning where real and complex parameterizations may differ, for example it does not treat implicit bias of gradient-based optimization [49]. Fourth, our experiments (Section 4) include only a single real-world setting. Finally, while we establish that a separation between real and complex parameterizations in terms of practical learnability takes place in three important cases (see Corollaries 1 to 3), these cases are likely far from being exhaustive. Indeed, Theorem 2 provides a mild sufficient condition for separation in terms of practical learnability—namely, that certain forward differences are not especially small—and we believe it is possible to apply analytical tools (*e.g.*, the Nørlund-Rice integral [17]) for translating this condition into interpretable properties satisfied in various important cases beyond those considered. Pursuing the latter direction, and more broadly, addressing the aforementioned limitations, are regarded as important directions for future work.

# 7 Discussion

The extent to which complex parameterizations benefit SSMs is an important open question in machine learning. Evidence in the literature is mixed: while some works demonstrate that complex parameterizations are essential for strong performance, others show that in various settings, real parameterizations lead to comparable (and in some cases better) performance. It was conjectured by Gu and Dao [20] that complex parameterizations are preferable for continuous data modalities (*e.g.*, audio, video), whereas for discrete data modalities (*e.g.*, text, DNA) real parameterizations suffice.

Since a complex SSM includes twice as many parameters as a real SSM of the same dimension, a priori, one might expect that a real SSM would benefit from becoming complex similarly to how it would benefit from doubling its dimension. Our theory showed that this is not the case, and in fact the former benefit far exceeds the latter. Indeed, we established separations between real and complex SSMs, by which a real SSM can only match a complex SSM if either the dimension of the real SSM or the number of iterations required for its training is exponentially large. Experiments corroborated our theory, and suggested that selectivity—a new architectural feature yielding state of the art performance—may be the dominant factor behind the aforementioned evidence in the literature being mixed.

We now outline a potential extension of our theory that accounts for selectivity. Roughly speaking, the separations we established between real and complex SSMs arise from a gap in their ability to express oscillations, *i.e.*, to express frequency components in their impulse response: while a complex SSM can easily express any frequency, a real SSM struggles to do so. Adding selectivity to a real SSM makes its parameters input-dependent, resulting in what can be viewed as an input-dependent impulse response. We hypothesize that this dependence allows importing frequency components from the input to the impulse response. If confirmed, this hypothesis would imply that when the input data is sufficiently rich in frequency content, selectivity can endow real SSMs with all the benefits we proved for complex SSMs. Such an outcome aligns with the conjecture of Gu and Dao [20]: continuous data modalities often consist of only low frequencies, whereas discrete data modalities typically have a "whiter spectrum," *i.e.*, a more uniform mix of frequencies [54].

We believe that extending our theory as described may elucidate the mixed evidence in the literature, thereby fully delineating the benefits of complex parameterizations for SSMs.

## Acknowledgments and Disclosure of Funding

We thank Itamar Zimerman for illuminating discussions. This work was supported the European Research Council (ERC) grants HOLI 819080 and NN4C 101164614, a Google Research Scholar Award, a Google Research Gift, Meta, the Yandex Initiative in Machine Learning, the Israel Science

Foundation (ISF) grant 1780/21, the Tel Aviv University Center for AI and Data Science, the Adelis Research Fund for Artificial Intelligence, Len Blavatnik and the Blavatnik Family Foundation, and Amnon and Anat Shashua.

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

# A Extensions of Theoretical Analysis

## A.1 Discretization

Various SSMs incorporate *discretization* [20, 21, 23, 16], which involves replacing parameter matrices with specific transformations that depend on an additional parameter $\Delta \in \mathbb{R}_{>0}$. With slight modifications, our main theoretical results, Theorem 1 and Theorem 2, apply directly to SSMs with common discretizations. Below, we exemplify this for one of the most common discretizations. Other discretizations can be treated similarly.

One of the most common discretizations is the bilinear discretization [21, 22], in which an SSM $(\overline{A}_\mathbb{K}, \overline{B}_\mathbb{K}, \overline{C}_\mathbb{K})$ is parameterized in the following way:

$$\overline{A}_\mathbb{K} = (I - \Delta/2 \cdot A_\mathbb{K})^{-1} (I + \Delta/2 \cdot A_\mathbb{K}), \quad \overline{B}_\mathbb{K} = (I - \Delta/2 \cdot A_\mathbb{K})^{-1} \Delta B_\mathbb{K}, \quad \overline{C}_\mathbb{K} = C_\mathbb{K},$$

where $\Delta \in \mathbb{R}_{>0}$, the matrices $A_\mathbb{K}, B_\mathbb{K}$ and $C_\mathbb{K}$ share the same dimensions and are defined over the same fields as $\overline{A}_\mathbb{K}, \overline{B}_\mathbb{K}$ and $\overline{C}_\mathbb{K}$, respectively, and the real parts of the diagonal entries of $A_\mathbb{K}$ are negative. Since the set of functions expressible by a real or complex SSM remains unchanged under this discretization, Theorem 1 holds without modification. Furthermore, Theorem 2 applies to this discretization if the left-hand side of Equation (3) is replaced by $n_\mathbb{R}\Delta\|C_\mathbb{R}^\top \odot B_\mathbb{R}\|_\infty$. This follows directly from Lemma 13, which shows that for any $i \in [n_\mathbb{R}]$, we have $|\Delta (B_\mathbb{R})_i| \geq |(\overline{B}_\mathbb{R})_i|$. The latter inequality justifies replacing $\overline{B}_\mathbb{R}$ with $\Delta B_\mathbb{R}$ on the left-hand side of Equation (3), leading to the desired result.

## A.2 Feedthrough

Some SSMs include an additional *feedthrough* parameter $D \in \mathbb{K}^{1,1}$ [41]. With feedthrough, the expression for $y(t)$ in Equation (1) becomes $\Re(C\mathbf{x}(t) + Du(t))$. Our theory essentially applies as is to SSMs with feedthrough, as explained below.

Obviously, an SSM of dimension $n$ without a feedthrough parameter can be emulated by an SSM of dimension $n$ with a feedthrough parameter $D$ by setting $D = 0$. Conversely, an SSM $(A, B, C)$ of dimension $n$ with a feedthrough parameter $D$ can be emulated by an SSM $(A', B', C')$ of dimension $n + 1$ without a feedthrough parameter. This is achieved, for example, by setting the diagonal entries of $A'$ to be those of $A$ followed by zero, the entries of $B'$ to be those of $B$ followed by one, and the entries of $C'$ to be those of $C$ followed by $D$. For any result where an SSM without feedthrough is assumed, a corresponding result for an SSM with feedthrough holds. Such a result can be attained by either emulating an SSM with feedthrough by an SSM without (while increasing dimension from $n$ to $n+1$) or by emulating an SSM without feedthrough by an SSM with feedthrough, (while maintaining the dimension) depending on the context.

## A.3 Complex $A$ with Real $B$ and $C$

It is possible to consider hybrid SSMs where $A$ is allowed to be complex while $B$ and $C$ are restricted to be real. Below we show that such hybrid SSMs enjoy all provable benefits of complex SSMs (*i.e.*, of SSMs where $A, B$ and $C$ can all be complex).

There are four results concerning complex SSMs: Proposition 1, Proposition 2, Theorem 1 and Proposition 3. The first three results can be trivially extended to the setting where only $A$ is complex, as their proofs do not assume complex $B$ or $C$. An extension of Proposition 3 is also straightforward. Indeed, the proof of Proposition 3 utilizes the fact that for a complex SSM $(A_\mathbb{C}, B_\mathbb{C}, C_\mathbb{C})$ of dimension $n_\mathbb{C}$, the impulse response can be any linear combination of $n_\mathbb{C}$ arbitrary decaying sine and cosine waves, which, by Fourier theory, can approximate any sequence of length $n_\mathbb{C}$. If only $A_\mathbb{C}$ is allowed to be complex, the impulse response can consist of any linear combination of $n_\mathbb{C}$ arbitrary decaying cosine waves, which can still represent any sequence of length $n_\mathbb{C}$ via the discrete cosine transform. Thus, the setting remains fundamentally similar, allowing for the extension of Proposition 3 to hybrid SSMs.

# B Exponentiality Impedes Practical Learning

In this appendix, we show that learning the parameters of the real SSM, *i.e.* $(A_\mathbb{R}, B_\mathbb{R}, C_\mathbb{R})$, is impractical when the required values for $(B_\mathbb{R}, C_\mathbb{R})$ are exponential in $t$. We account for two aspects of this impracticality: the number of iterations required by gradient descent; and the precision (number

of bits) required for representing the parameters. Note that we will not preclude the possibility of overcoming the impracticality through development of new techniques (*e.g.* new parameterizations for $B_\mathbb{R}$ and $C_\mathbb{R}$). We believe our account herein may assist in such developments.

**Exponential number of iterations.**   For the sake of illustration, suppose that training the real SSM to realize a mapping $\phi_{n_\mathbb{R},(A_\mathbb{R},B_\mathbb{R},C_\mathbb{R})}(\cdot)$ that $\epsilon$-approximates $\phi(\cdot)$ up to time $t$, is implemented via gradient descent over a loss function $\ell(\cdot)$ that measures the squared error of the output at time $t$, where input elements are drawn independently from the standard normal distribution:

$$\ell(A_\mathbb{R},B_\mathbb{R},C_\mathbb{R}) := \mathbb{E}_{\mathbf{U}\in\mathbb{R}^\mathbb{N},\ \mathbf{U}_1,\mathbf{U}_2,\dots \overset{\text{i.i.d.}}{\sim} \mathcal{N}(0,1)} \left[ (\phi_{n_\mathbb{R},(A_\mathbb{R},B_\mathbb{R},C_\mathbb{R})}(\mathbf{U})_t - \phi(\mathbf{U})_t)^2 \right]. \qquad (7)$$

By a simple derivation (see Appendix C) $\ell(A_\mathbb{R},B_\mathbb{R},C_\mathbb{R}) = \|\phi_{n_\mathbb{R},(A_\mathbb{R},B_\mathbb{R},C_\mathbb{R})}(\mathbf{I})_{:t} - \phi(\mathbf{I})_{:t}\|_2^2$, therefore sufficient minimization of the loss $\ell(\cdot)$ (namely, minimization of $\ell(\cdot)$ to or below the value $\epsilon^2/t$) indeed guarantees that $\phi_{n_\mathbb{R},(A_\mathbb{R},B_\mathbb{R},C_\mathbb{R})}(\cdot)$ $\epsilon$-approximates $\phi(\cdot)$ up to time $t$. Assume the following regularity conditions on gradient descent: *(i)* the step sizes (learning rates) for all iterations are upper bounded by a constant (independent of $t$); *(ii)* the step size for each iteration is *stable*, in the sense that it is upper bounded by $2/\lambda_{max}$, where $\lambda_{max}$ represents the maximum eigenvalue of the Hessian of $\ell(\cdot)$ at the respective iteration [13, 7, 15]; and *(iii)* the values of $\ell(\cdot)$ throughout all iterations are upper bounded by a constant (independent of $t$) times the value of $\ell(\cdot)$ at initialization. Proposition 4 below establishes that under said regularity conditions, during gradient descent, the magnitude of the parameters $(B_\mathbb{R},C_\mathbb{R})$ grows at most linearly in the number of iterations. Accordingly, if the required values for $(B_\mathbb{R},C_\mathbb{R})$ are exponential in $t$ and the initialization of $(B_\mathbb{R},C_\mathbb{R})$ is not, attaining the required values necessitates an exponential (in $t$) number of iterations.

**Proposition 4.** *Consider an application of gradient descent to the loss $\ell(\cdot)$ in Equation* (7)*:*

$$\left(A_\mathbb{R}^{(i)},B_\mathbb{R}^{(i)},C_\mathbb{R}^{(i)}\right) = \left(A_\mathbb{R}^{(i-1)},B_\mathbb{R}^{(i-1)},C_\mathbb{R}^{(i-1)}\right) - \eta^{(i)}\nabla\ell\left(A_\mathbb{R}^{(i-1)},B_\mathbb{R}^{(i-1)},C_\mathbb{R}^{(i-1)}\right),\ i\in\mathbb{N},$$

*where $\left(A_\mathbb{R}^{(0)},B_\mathbb{R}^{(0)},C_\mathbb{R}^{(0)}\right)$ is a chosen initialization, and $\eta^{(i)}\in\mathbb{R}_{>0}$ represents the step size selected for iteration $i\in\mathbb{N}$. Assume $\exists c_1\in\mathbb{R}_{>0}$ such that $\forall i\in\mathbb{N}:\eta^{(i)}\le c_1$. Assume also:*

$$\forall i\in\mathbb{N}:\eta^{(i)}\le 2\Big/\max\left\{\lambda_{max}\left(\nabla^2\ell\left(A_\mathbb{R}^{(i-1)},B_\mathbb{R}^{(i-1)},C_\mathbb{R}^{(i-1)}\right)\right),0\right\},$$

*where $\lambda_{max}(M)$, for a symmetric matrix $M$, is the maximum eigenvalue of $M$. Finally, assume:*

$$\exists c_2\in\mathbb{R}_{>0}\ \text{ such that }\ \forall i\in\mathbb{N}:\ell\left(A_\mathbb{R}^{(i)},B_\mathbb{R}^{(i)},C_\mathbb{R}^{(i)}\right)\le c_2\cdot\ell\left(A_\mathbb{R}^{(0)},B_\mathbb{R}^{(0)},C_\mathbb{R}^{(0)}\right).$$

*Then:*

$$\forall i\in\mathbb{N}:\max\left\{\left\|B_\mathbb{R}^{(i)}\right\|_\infty,\left\|(C_\mathbb{R}^{(i)})^\top\right\|_\infty\right\}\le\max\left\{\left\|B_\mathbb{R}^{(0)}\right\|_\infty,\left\|(C_\mathbb{R}^{(0)})^\top\right\|_\infty\right\}$$
$$+\ i\cdot\left(4c_1c_2t\cdot\ell\left(A_\mathbb{R}^{(0)},B_\mathbb{R}^{(0)},C_\mathbb{R}^{(0)}\right)\right)^{0.5}.$$

*Proof sketch (proof in Appendix D.8).* Taking into account the form of $\ell(\cdot)$ (see Appendix C) and the assumptions made, the proof shows that at each iteration of gradient descent, each entry in $B_\mathbb{R}$ or $C_\mathbb{R}$ changes by at most $(4c_1c_2t\cdot\ell(A_\mathbb{R}^{(0)},B_\mathbb{R}^{(0)},C_\mathbb{R}^{(0)}))^{0.5}$. This readily leads to the desired result. $\qquad\square$

**Prohibitive precision.**   With conventional floating-point representation, a real number $\rho$ is represented as $s\cdot k\cdot 2^m$, where: $s\in\{-1,1\}$ (the sign) is represented by one bit; $k\in\mathbb{N}\cup\{0\}$ (the significand) is represented by $b_k\in\mathbb{N}$ bits; and $m\in\mathbb{Z}$ (the exponent) is represented by $b_m\in\mathbb{N}$ bits. The precision of the floating-point representation is the total number of bits used for $s$, $k$ and $m$, *i.e.* it is $1+b_k+b_m$. For example, with the widespread IEEE 754 standard [26]: single precision corresponds to 32 bits with $b_k=23, b_m=8$; and double precision corresponds to 64 bits with $b_k=52, b_m=11$. It is customary to model quantization error as an additive random variable uniformly distributed over the interval $[-\xi/2,\xi/2]$, where $\xi\in\mathbb{R}_{>0}$ is the quantization step size [56]. With the floating-point representation of $\rho$, the best (lowest) achievable quantization step size is on the order of $|\rho|\cdot 2^{-b_k}$, thus we may model quantization error as a multiplicative random variable uniformly distributed over $[1-q/2,1+q/2]$, where $q=\Theta(2^{-b_k})$. Definition 2 below employs this model to formalize the notion of robustness to quantization for values of $(A_\mathbb{R},B_\mathbb{R},C_\mathbb{R})$ with which $\phi_{n_\mathbb{R},(A_\mathbb{R},B_\mathbb{R},C_\mathbb{R})}(\cdot)$ $\epsilon$-approximates $\phi(\cdot)$ up to time $t$. Specifically, Definition 2 defines *robustness to q-quantization* for such values to be the probability that $\phi_{n_\mathbb{R},(A_\mathbb{R},B_\mathbb{R},C_\mathbb{R})}(\cdot)$ continues

to $\epsilon$-approximate $\phi(\cdot)$ up to time $t$, when the values are multiplied by random variables uniformly distributed over $[1 - q/2, 1 + q/2]$. Proposition 5 below then proves that robustness to $q$-quantization is (at most) inversely proportional to $q$ times the magnitude of $(B_\mathbb{R}, C_\mathbb{R})$. Recalling that $q = \Theta(2^{-b_k})$, we conclude that if the values of $(B_\mathbb{R}, C_\mathbb{R})$ are exponential in $t$, a non-negligible robustness to quantization necessitates having $b_k = \Omega(t)$, meaning a floating-point precision that scales (at least) linearly in $t$. The latter requirement is prohibitive, since virtually all computing systems implementing neural networks entail fixed options for floating-point precision (typically 16, 32 and 64 [1, 43]), all much smaller than what the time $t$ can be (tens of thousands or more [20, 47, 19, 8])

**Definition 2.** Let $q \in [0, 1]$, and let $Q_{A_\mathbb{R}}$, $Q_{B_\mathbb{R}}$ and $Q_{C_\mathbb{R}}$ be random matrices of the same sizes as $A_\mathbb{R}$, $B_\mathbb{R}$ and $C_\mathbb{R}$, respectively, wherein each entry is independently drawn from the uniform distribution over $[1 - q/2, 1 + q/2]$. Let $(A_\mathbb{R}^*, B_\mathbb{R}^*, C_\mathbb{R}^*)$ be values for $(A_\mathbb{R}, B_\mathbb{R}, C_\mathbb{R})$ with which $\phi_{n_\mathbb{R}, (A_\mathbb{R}, B_\mathbb{R}, C_\mathbb{R})}(\cdot)$ $\epsilon$-approximates $\phi(\cdot)$ up to time $t$. The *robustness to $q$-quantization* of $(A_\mathbb{R}^*, B_\mathbb{R}^*, C_\mathbb{R}^*)$ is the probability that $\phi_{n_\mathbb{R}, (A_\mathbb{R}, B_\mathbb{R}, C_\mathbb{R})}(\cdot)$ $\epsilon$-approximates $\phi(\cdot)$ up to time $t$ when $(A_\mathbb{R}, B_\mathbb{R}, C_\mathbb{R})$ take the values $(A_\mathbb{R}^* \odot Q_{A_\mathbb{R}}, B_\mathbb{R}^* \odot Q_{B_\mathbb{R}}, C_\mathbb{R}^* \odot Q_{C_\mathbb{R}})$.

**Proposition 5.** *Suppose that $\phi_{n_\mathbb{R}, (A_\mathbb{R}, B_\mathbb{R}, C_\mathbb{R})}(\cdot)$ $\epsilon$-approximates $\phi(\cdot)$ up to time $t$. Then, for any $q \in [0, 1]$, the robustness to $q$-quantization of the values held by $(A_\mathbb{R}, B_\mathbb{R}, C_\mathbb{R})$ is at most:*

$$2\epsilon / \left( q \| C_\mathbb{R}^\top \odot B_\mathbb{R} \|_\infty \right).$$

*Proof sketch (proof in Appendix D.9).* Denote by $(A_\mathbb{R}^*, B_\mathbb{R}^*, C_\mathbb{R}^*)$ the values held by $(A_\mathbb{R}, B_\mathbb{R}, C_\mathbb{R})$, and let $Q_{A_\mathbb{R}}$, $Q_{B_\mathbb{R}}$ and $Q_{C_\mathbb{R}}$ be random matrices as in Definition 2. When $(A_\mathbb{R}, B_\mathbb{R}, C_\mathbb{R})$ take the values $(A_\mathbb{R}^* \odot Q_{A_\mathbb{R}}, B_\mathbb{R}^* \odot Q_{B_\mathbb{R}}, C_\mathbb{R}^* \odot Q_{C_\mathbb{R}})$, the first entry of the impulse response of $\phi_{n_\mathbb{R}, (A_\mathbb{R}, B_\mathbb{R}, C_\mathbb{R})}(\cdot)$ equals $(C_\mathbb{R}^* \odot Q_{C_\mathbb{R}})(B_\mathbb{R}^* \odot Q_{B_\mathbb{R}})$. It suffices to restrict attention to this first entry, and show that $|(C_\mathbb{R}^* \odot Q_{C_\mathbb{R}})(B_\mathbb{R}^* \odot Q_{B_\mathbb{R}}) - \phi(\mathbf{I})_1| \leq \epsilon$ with probability at most $2\epsilon / (q \| C_\mathbb{R}^\top \odot B_\mathbb{R} \|_\infty)$. The proof establishes the latter anti-concentration result. $\square$

## C  Loss Functions

Consider a loss function as in our theory (see Equation (7)), applied to both real and complex SSMs:

$$\ell(A_\mathbb{K}, B_\mathbb{K}, C_\mathbb{K}) = \mathbb{E}_{\mathbf{U} \in \mathbb{R}^\mathbb{N}, \, \mathbf{U}_1, \mathbf{U}_2, \dots \overset{\text{i.i.d.}}{\sim} \mathcal{N}(0,1)} \left[ \left( \phi_{n_\mathbb{K}, (A_\mathbb{K}, B_\mathbb{K}, C_\mathbb{K})}(\mathbf{U})_t - \phi(\mathbf{U})_t \right)^2 \right],$$

where $\mathbb{K} = \mathbb{R}$ or $\mathbb{K} = \mathbb{C}$. Below we prove that:

$$\ell(A_\mathbb{K}, B_\mathbb{K}, C_\mathbb{K}) = \left\| \phi_{n_\mathbb{K}, (A_\mathbb{K}, B_\mathbb{K}, C_\mathbb{K})}(\mathbf{I})_{:t} - \phi(\mathbf{I})_{:t} \right\|_2^2.$$

Indeed, as discussed in Section 2.3, for any $\mathbf{U} \in \mathbb{R}^\mathbb{N}$:

$$\phi_{n_\mathbb{K}, (A_\mathbb{K}, B_\mathbb{K}, C_\mathbb{K})}(\mathbf{U})_t = \left( \mathbf{U} * \phi_{n_\mathbb{K}, (A_\mathbb{K}, B_\mathbb{K}, C_\mathbb{K})}(\mathbf{I}) \right)_t, \quad \phi(\mathbf{U})_t = (\mathbf{U} * \phi(\mathbf{I}))_t.$$

Thus:

$$\phi_{n_\mathbb{K}, (A_\mathbb{K}, B_\mathbb{K}, C_\mathbb{K})}(\mathbf{U})_t - \phi(\mathbf{U})_t = \sum_{j=1}^{t} \mathbf{U}_j \left( \phi_{n_\mathbb{K}, (A_\mathbb{K}, B_\mathbb{K}, C_\mathbb{K})}(\mathbf{I})_{t-j+1} - \phi(\mathbf{I})_{t-j+1} \right).$$

It holds that:

$$\ell(A_\mathbb{R}, B_\mathbb{R}, C_\mathbb{R}) = \mathbb{E}_{\mathbf{U} \in \mathbb{R}^\mathbb{N}, \, \mathbf{U}_1, \mathbf{U}_2, \dots \overset{\text{i.i.d.}}{\sim} \mathcal{N}(0,1)} \left[ \left( \sum_{j=1}^{t} \mathbf{U}_j \left( \phi_{n_\mathbb{K}, (A_\mathbb{K}, B_\mathbb{K}, C_\mathbb{K})}(\mathbf{I})_{t-j+1} - \phi(\mathbf{I})_{t-j+1} \right) \right)^2 \right]$$

$$= \sum_{j=1}^{t} \left( \phi_{n_\mathbb{K}, (A_\mathbb{K}, B_\mathbb{K}, C_\mathbb{K})}(\mathbf{I})_{t-j+1} - \phi(\mathbf{I})_{t-j+1} \right)^2 \mathbb{E}_{\mathbf{U}_j \sim \mathcal{N}(0,1)} \left[ \mathbf{U}_j^2 \right]$$

$$= \left\| \phi_{n_\mathbb{K}, (A_\mathbb{K}, B_\mathbb{K}, C_\mathbb{K})}(\mathbf{I})_{:t} - \phi(\mathbf{I})_{:t} \right\|_2^2,$$

as required.

## D  Deferred Proofs

### D.1  Proof of Proposition 1

Beginning with the real SSM ($\mathbb{K} = \mathbb{R}$), assume that $n_\mathbb{R} \geq t$. We would like to show that there exist assignments for $(A_\mathbb{R}, B_\mathbb{R}, C_\mathbb{R})$ with which:

$$\phi(\mathbf{I})_{:t} = (C_\mathbb{R} B_\mathbb{R}, C_\mathbb{R} A_\mathbb{R} B_\mathbb{R}, C_\mathbb{R} A_\mathbb{R}^2 B_\mathbb{R}, \ldots, C_\mathbb{R} A_\mathbb{R}^{t-1} B_\mathbb{R}),$$

where $\phi(\mathbf{I})_{:t}$ stands for the first $t$ elements of the impulse response of $\phi(\cdot)$. Regarding $\phi(\mathbf{I})_{:t}$, $B_\mathbb{R}$ and $C_\mathbb{R}^\top$ as (column) vectors, we may write the sought after equality as $V(A_\mathbb{R})(C_\mathbb{R}^\top \odot B_\mathbb{R}) = \phi(\mathbf{I})_{:t}$, where $V(A_\mathbb{R}) \in \mathbb{R}^{t,n_\mathbb{R}}$ is a matrix whose $(i,j)$th entry holds the $j$th diagonal entry of $A_\mathbb{R}$ exponentiated by $i-1$. Notice that $V(A_\mathbb{R})$ is a Vandermonde matrix comprising powers of the diagonal entries of $A_\mathbb{R}$. Any assignment for $A_\mathbb{R}$ with distinct diagonal entries leads the Vandermonde matrix to have full rank, which in turn means that there exists a vector $\mathbf{v} \in \mathbb{R}^{n_\mathbb{R}}$ satisfying $V(A_\mathbb{R})\mathbf{v} = \phi(\mathbf{I})_{:t}$. Assigning $B_\mathbb{R} = \mathbf{v}$ and $C_\mathbb{R}^\top = \mathbf{1}$ concludes the proof for the real SSM. The complex SSM ($\mathbb{K} = \mathbb{C}$) can be treated analogously, while assigning real values to $A_\mathbb{C}$, $B_\mathbb{C}$ and $C_\mathbb{C}$ in order to allow disregarding the fact that only real parts of outputs are taken (*i.e.*, disregarding the operator $\Re(\cdot)$ in Equation (2)). $\qquad\square$

### D.2  Proof of Theorem 1

To facilitate the proof of the theorem, we first outline essential definitions and lemmas to quanitify a lower bound for the number of times the impulse response alternates between being greater than or equal to $1/4$, and being smaller than or equal to $-1/4$. Building on this groundwork, we then unpack the main theorem's proof in Appendix D.2.2.

#### D.2.1  Setup for the proof

**Definition 3.**  For an angle $\theta$ in $\mathbb{R}$ we will say that $\theta$ is in the **mostly-positive region** if

$$\theta \bmod 2\pi \in \left[0, \frac{\pi}{3}\right] \cup \left[\frac{5\pi}{3}, 2\pi\right]$$

and we say that $\theta$ is in the **mostly-negative region** if

$$\theta \bmod 2\pi \in \left[\frac{2\pi}{3}, \frac{4\pi}{3}\right]$$

**Remark 1.**  *Under this definition, a non-zero complex number $x$ has that its argument is in the mostly-positive region if and only if $\Re(x) \geq \frac{|x|}{2}$. Conversely, $x$ has that its argument is in the mostly-negative region if and only if $\Re(x) \leq -\frac{|x|}{2}$.*

**Definition 4.**  We say that a set is **balanced** if at least a sixth of its elements are in the mostly-positive region and at least a sixth of its elements are in the mostly-negative region.

**Definition 5.**  The **balancing number** of a complex number $x$ with argument $\theta$ is defined as the minimum $p \in \mathbb{N}$ such that for any non-zero real number $b$ the set $\{b + \theta, b + 2\theta, \ldots, b + p\theta\}$ is balanced. If no such $p$ exists, we say that the balancing number of $x$ is infinity. We denote the balancing number of $x$ by $\beta(x)$.

We will begin by establishing an upper bound on $\beta(x)$ for $x$ in a specific region in the complex plane. Subsequently, we generalize this result to give an upper bound of the balancing number for almost all complex numbers.

**Lemma 1.**  *For any natural number $p \geq 4$, if a non-zero complex number $x$ has an argument $\theta$ such that $\theta$ is in $[\frac{2\pi}{p}, \frac{2\pi}{3}]$, then $\beta(x) \leq p$. Moreover, for any natural number $p \geq 9$, if $x$ has an argument $\theta$ such that $\theta$ is in $[\pi + \frac{2\pi}{p-1}, \frac{4\pi}{3}]$ then $\beta(x) \leq p$.*

*Proof.*  Since the mostly-positive and mostly-negative regions are defined by looking at the angle of a complex number mod $2\pi$, to prove that $\beta(x) \leq p$, it is enough to show there exists some number $k \leq p$ such that for any real $b$ the following set

$$\{(b + \theta) \bmod 2\pi, (b + 2 \cdot \theta) \bmod 2\pi, \ldots, (b + k \cdot \theta) \bmod 2\pi\}$$

is balanced.

Indeed, let there be some real $b$.

1. If $\theta$ is in $[\frac{2\pi}{p}, \frac{2\pi}{3}]$ and $p \geq 4$, Let $k$ be the natural number such that $2\pi < \theta k \leq 2\pi + \theta$. It follows that $4 \leq k \leq p$ and that $\theta \leq \frac{2\pi}{k-1}$. By Lemma 5 we have that the set

$$\{(b+\theta) \bmod 2\pi, (b+2\cdot\theta) \bmod 2\pi, \ldots, (b+k\cdot\theta) \bmod 2\pi\}$$

   has at least a single point in any continuous interval (in mod $2\pi$) of length $\theta$. As such, any continuous interval mod $2\pi$ of length $\frac{2\pi}{3}$ must contain at least

$$\left\lfloor \frac{\frac{2\pi}{3}}{\theta} \right\rfloor \geq \left\lfloor \frac{\frac{2\pi}{3}}{\frac{\pi}{k-1}} \right\rfloor = \left\lfloor \frac{k-1}{3} \right\rfloor \geq \frac{k}{6}$$

   points, where the last inequality is true for any $k \geq 4$. Since the mostly-positive and mostly-negative regions are continuous intervals (mod $2\pi$) of length $\frac{2\pi}{3}$, we have that the set

$$\{(b+\theta) \bmod 2\pi, (b+2\cdot\theta) \bmod 2\pi, \ldots, (b+k\cdot\theta) \bmod 2\pi\}$$

   has at least a sixth of its points in the mostly-positive and mostly-negative regions, and is therefore balanced, as needed.

2. If $\theta$ is in $[\frac{\pi}{2} + \frac{2\pi}{p-1}, \frac{4\pi}{3}]$ and $p \geq 9$, then $2\theta \bmod \pi$ is in the interval $\left[\frac{2\pi}{\frac{p-1}{2}}, \frac{2\pi}{3}\right]$ which is contained in the interval $\left[\frac{2\pi}{\lfloor\frac{p}{2}\rfloor}, \frac{2\pi}{3}\right]$. Therefore, by the previous section, there exists some $k \leq \lfloor\frac{p}{2}\rfloor$ such that the following set:

$$\{b + 2\theta \bmod 2\pi, \ldots, b + k\cdot 2\theta \bmod 2\pi\}$$

   and the set

$$\{(b-\theta) + 2\theta \bmod 2\pi, \ldots, (b-\theta) + k\cdot 2\theta \bmod 2\pi\}$$

   are both balanced. Overall, their union, the set

$$\{(b+\theta) \bmod 2\pi, (b+2\cdot\theta) \bmod 2\pi, \ldots, (b+2k\cdot\theta) \bmod 2\pi\}$$

   is also balanced, showing that the $\beta(x) \leq 2k \leq p$ as needed.

$\square$

**Lemma 2.** *For any natrual numbers $r, \theta$ the following holds:*

$$\beta\left(e^{r+i\theta}\right) = \beta\left(e^{r-i\theta}\right)$$

*Proof.* For any rational numbers $\theta$ and $b$, and for any natural number $p$, we will define the set $S_{\theta,b,p}$ in the following way:

$$S_{\theta,b,p} = \{(b+\theta) \bmod 2\pi, (b+2\cdot\theta) \bmod 2\pi, \ldots, (b+p\cdot\theta) \bmod 2\pi\}$$

We will also define the set $S_{\theta,p}$ as

$$S_{\theta,p} = \{S_{\theta,b,p} | b \in \mathbb{C}\}.$$

Given this notation, we have that $\beta\left(e^{r-i\theta}\right) = p$ if and only if $p$ is the minimum natural number such that every set in $S_{\theta,p}$ is balanced. For any rational numbers $\theta$ and $b$ and for any natural number $p$, we have

$$S_{-\theta,-b,p} = -S_{\theta,b,p}$$

(Where minus is taken element-wise mod $2\pi$). This immediately gives:

$$S_{-\theta,p} = -S_{\theta,p}.$$

Since the mostly-positive and mostly-negative regions are invariant to taking a minus mod $2\pi$, we have that all the sets in $S_{\theta,p}$ are balanced if and only if so are all the sets of $S_{-\theta,p}$. This directly implies that $\beta\left(e^{r+i\theta}\right) = \beta\left(e^{r-i\theta}\right)$ as needed. $\square$

**Corollary 4.** *If $x \in \mathbb{C}$ has that $\arg(x) \bmod \pi \in [\frac{2\pi}{p-1}, \pi - \frac{2\pi}{p-1}]$ for some $p \geq 9$ then $\beta(x) \leq p$*

*Proof.* Denote $\theta \equiv \arg(x)$. From Lemma 1 if the angle $\theta$ lies within the interval

$$I = \left[\frac{2\pi}{p}, \frac{2\pi}{3}\right] \cup \left[\pi + \frac{2\pi}{p-1}, \frac{4\pi}{3}\right]$$

then $\beta(x) \le p$. Additionally, Lemma 2 indicates that the same holds if $\theta$ is in the interval $-I$ (where operations are element-wise mod $2\pi$). Hence, we have that if $\theta$ is contained in interval $I'$, where

$$\begin{aligned}
I' &= I \cup (-I) \\
&= \left[\frac{2\pi}{p}, \frac{2\pi}{3}\right] \cup \left[\pi + \frac{2\pi}{p-1}, \frac{4\pi}{3}\right] \cup \left[\frac{4\pi}{3}, 2\pi - \frac{2\pi}{p}\right] \cup \left[\frac{2\pi}{3}, \pi - \frac{2\pi}{p-1}\right] \\
&= \left[\frac{2\pi}{p}, \pi - \frac{2\pi}{p-1}\right] \cup \left[\pi + \frac{2\pi}{p-1}, 2\pi - \frac{2\pi}{p}\right]
\end{aligned}$$

then $\beta(x) \le p$. Overall, this means that if $\theta$ has that

$$\theta \bmod \pi \in \left[\frac{2\pi}{p-1}, \pi - \frac{2\pi}{p-1}\right]$$

then $\beta(x) \le p$, which gives the Corollary. $\qquad\square$

### D.2.2 The Proof

Let $\epsilon > 0$ be any positive number. For any sequence $X = \{X_1, X_2, \ldots, X_t\}$, let $X|_1$ denote the subsequence consisting of elements at odd indices, i.e., $X|_1 = \{X_1, X_3, \ldots\}$.

First, we notice that from the assumption that $|\sin(\arg(A_\mathbb{C}))| \ge 0.2$ we have that $\arg(A_\mathbb{C}) \in \left[\frac{\pi}{16}, \pi - \frac{\pi}{16}\right]$, and so

$$\arg(A_\mathbb{C}^2) \bmod \pi = 2\arg(A_\mathbb{C}) \bmod \pi \in 2 \cdot \left[\frac{\pi}{16}, \pi - \frac{\pi}{16}\right] \bmod \pi = \left[\frac{2\pi}{9-1}, \pi - \frac{2\pi}{9-1}\cdot\right]$$

This implies, by Corollary 4, that the balancing number of $A_\mathbb{C}^2$ is less than 9. Now, if $\phi_{n_\mathbb{R}, (A_\mathbb{R}, B_\mathbb{R}, C_\mathbb{R})}(\cdot)$ $\epsilon$-approximates $\phi_{n_\mathbb{C}, (A_\mathbb{C}, B_\mathbb{C}, C_\mathbb{C})}(\cdot)$ up to time $t$, then, by definition,

$$\|\phi_{n_\mathbb{R}, (A_\mathbb{R}, B_\mathbb{R}, C_\mathbb{R})}(\mathbf{I})_{:t} - \phi_{n_\mathbb{C}, (A_\mathbb{C}, B_\mathbb{C}, C_\mathbb{C})}(\mathbf{I})_{:t}\|_1 \le \epsilon.$$

This implies

$$\|\phi_{n_\mathbb{R}, (A_\mathbb{R}, B_\mathbb{R}, C_\mathbb{R})}(\mathbf{I})_{:t}|_1 - \phi_{n_\mathbb{C}, (A_\mathbb{C}, B_\mathbb{C}, C_\mathbb{C})}(\mathbf{I})_{:t}|_1\|_1 \le \epsilon.$$

Now, we have that

$$\begin{aligned}
\left(\phi_{1, (A_\mathbb{C}, B_\mathbb{C}, C_\mathbb{C})}(\mathbf{I})_{:t}|_1\right)_k &= \left(\phi_{1, (A_\mathbb{C}, B_\mathbb{C}, C_\mathbb{C})}(\mathbf{I})_{:t}\right)_{2k-1} \\
&= \Re\left(B_\mathbb{C} C_\mathbb{C} A_\mathbb{C}^{2k-2}\right)
\end{aligned}$$

and because the balancing of $A_\mathbb{C}^2$ is at most 9, this implies that there exists a subsequence of $1, 2, \ldots, \lfloor t/2 \rfloor$ with increasing indices $k_j$ of length at least $\lfloor \frac{t}{9} \rfloor - 1$ such that if $j$ is even then $B_\mathbb{C} C_\mathbb{C} \left(A_\mathbb{C}^2\right)^{k_j - 1}$ is in the mostly-positive region, which implies

$$\Re\left(B_\mathbb{C} C_\mathbb{C} \left(A_\mathbb{C}^2\right)^{k_j - 1}\right) \ge |B_\mathbb{C} C_\mathbb{C}| \left|\frac{\left(A_\mathbb{C}^2\right)^{k_j - 1}}{2}\right| \ge \frac{1}{4},$$

where the first inequality is due to the definition of the mostly-positive region and the second is implied from the assumption that $|B_\mathbb{C} \cdot C_\mathbb{C}| \ge 1$ and $|A_\mathbb{C}| \ge 0.5^{1/t}$. Conversely, if $j$ is odd it is in the mostly-negative region, which implies

$$\Re\left(B_\mathbb{C} C_\mathbb{C} \left(A_\mathbb{C}^2\right)^{k_j - 1}\right) \le -|B_\mathbb{C} C_\mathbb{C}| \left|\frac{\left(A_\mathbb{C}^2\right)^{k_j - 1}}{2}\right| \le -\frac{1}{4}.$$

Next, we will show that the impulse response of any real system, when restricted to even indices, can change its sign at most $n_\mathbb{R} - 1$ times. This will prove the theorem since

$$\frac{1}{4}\left(\left\lfloor \frac{t}{9} \right\rfloor - n_\mathbb{R} - 1\right) \le \|\phi_{n_\mathbb{R}, (A_\mathbb{R}, B_\mathbb{R}, C_\mathbb{R})}(\mathbf{I})_{:t}|_1 - \phi_{n_\mathbb{C}, (A_\mathbb{C}, B_\mathbb{C}, C_\mathbb{C})}(\mathbf{I})_{:t}|_1\|_1 \le \epsilon.$$

The first inequality results from the fact that there would be at least $\left\lfloor \frac{t}{p} \right\rfloor - n_{\mathbb{R}} - 1$ indices where $\frac{1}{4} \leq \left| \left( \phi_{1,(A_{\mathbb{C}},B_{\mathbb{C}},C_{\mathbb{C}})} \left( \mathbf{I} \right)_{:t} |_1 \right)_k \right|$ and $\phi_{n_{\mathbb{R}},(A_{\mathbb{R}},B_{\mathbb{R}},C_{\mathbb{R}})} \left( \mathbf{I} \right)_{:t} |_1$ and $\phi_{1,(A_{\mathbb{C}},B_{\mathbb{C}},C_{\mathbb{C}})} \left( \mathbf{I} \right)_{:t} |_1$ disagree on sign, which yields the theorem.

Indeed, let us upper bound the number of sign changes of the real impulse response when restricted to odd indices. We have that

$$\left( \phi_{n_{\mathbb{R}},(A_{\mathbb{R}},B_{\mathbb{R}},C_{\mathbb{R}})} \left( \mathbf{I} \right)_{:t} |_1 \right)_k = \left( \phi_{n_{\mathbb{R}},(A_{\mathbb{R}},B_{\mathbb{R}},C_{\mathbb{R}})} \left( \mathbf{I} \right)_{:t} \right)_{2k-1}$$
$$= \Re \left( C_{\mathbb{R}} A_{\mathbb{R}}^{2k-2} B_{\mathbb{R}} \right)$$
$$= C_{\mathbb{R}} A_{\mathbb{R}}^{2k-2} B_{\mathbb{R}}$$
$$= \sum_{j=1}^{n_{\mathbb{R}}} (B_{\mathbb{R}})_j (C_{\mathbb{R}})_j (A_{\mathbb{R}})_{j,j}^{2k-2} .$$

By the mean value theorem, the number of times $\phi_{n_{\mathbb{R}},(A_{\mathbb{R}},B_{\mathbb{R}},C_{\mathbb{R}})} \left( \mathbf{I} \right)_{:t} |_1$ changes signs is bounded by the number of zeros of the following continuous function:

$$f(x) = \sum_{j=1}^{n_{\mathbb{R}}} (B_{\mathbb{R}})_j (C_{\mathbb{R}})_j \left( (A_{\mathbb{R}})_{j,j}^2 \right)^x .$$

This function is a linear combination of $n_{\mathbb{R}}$ exponential functions with a positive base, and so by Lemma 4, it has at most $n_{\mathbb{R}} - 1$ zeros, as needed.

$\square$

### D.3  Proof of Theorem 2

Let $d, m$ be natural numbers such that $d + m \leq \lfloor t/2 \rfloor$, and let $\sigma$ be in {odd, even}. We will show that

$$n_{\mathbb{R}} \| C_{\mathbb{R}}^\top \odot B_{\mathbb{R}} \|_\infty \geq 2^{d+2\min\{d,m\}} \left( 2^{-d} \left| (\phi \left( \mathbf{I} \right) |_\sigma)_m^{(d)} \right| - \epsilon \right) ,$$

thus proving the theorem.

Let $\phi_{n_{\mathbb{R}},(A_{\mathbb{R}},B_{\mathbb{R}},C_{\mathbb{R}})} (\cdot)$ be a real system that $\epsilon$-approximates $\phi (\cdot)$ up to time $t$. For convenience, we will denote $\phi_{n_{\mathbb{R}},(A_{\mathbb{R}},B_{\mathbb{R}},C_{\mathbb{R}})} \left( \mathbf{I} \right)_{:t}$ by $\mathbf{Y}$ and $\phi \left( \mathbf{I} \right)_{:t}$ by $\mathbf{T}$. Then, by definition, we have that

$$\| \mathbf{T} - \mathbf{Y} \|_1 \leq \epsilon,$$

which implies that

$$| \mathbf{T} |_\sigma - \mathbf{Y} |_\sigma |_\infty \leq \epsilon.$$

Lemma 9 guarantees that there exist some parameters $\left( \tilde{A}_{\mathbb{R}}, \tilde{B}_{\mathbb{R}}, \tilde{C}_{\mathbb{R}} \right)$ where $\tilde{A}_{\mathbb{R}}$ is non-negative and the following two things hold:

$$\| C_{\mathbb{R}}^T \odot B_{\mathbb{R}} \|_\infty \geq \| \tilde{C}_{\mathbb{R}}^T \odot \tilde{B}_{\mathbb{R}} \|_\infty$$
$$\mathbf{Y} |_\sigma = \phi_{n_{\mathbb{R}},\left( \tilde{A}_{\mathbb{R}}, \tilde{B}_{\mathbb{R}}, \tilde{C}_{\mathbb{R}} \right)} \left( \mathbf{I} \right)_{:t} .$$

Because $\tilde{A}_{\mathbb{R}}$ is non-negative, $\mathbf{Y} |_\sigma$ can be viewed as a linear combination of decaying exponentials with positive coefficients. Since the forward difference is linear, Lemma 6 ensures that the absolute value of the $d$th forward difference of $\mathbf{Y} |_\sigma$ at index $m$ is upper bounded by

$$\left( \frac{m}{d+m} \right)^m \left( \frac{d}{d+m} \right)^d \cdot n_{\mathbb{R}} \| \tilde{C}_{\mathbb{R}}^T \odot \tilde{B}_{\mathbb{R}} \|_\infty .$$

Using Lemma 12, we get

$$\left| (\mathbf{Y} |_\sigma)_m^{(d)} \right| \leq \frac{n_{\mathbb{R}} \| \tilde{C}_{\mathbb{R}}^T \odot \tilde{B}_{\mathbb{R}} \|_\infty}{2^{2\min(m,d)}} .$$

Lemma 8 implies that

$$\||\mathbf{T}|_\sigma - \mathbf{Y}|_\sigma\|_\infty \geq \frac{\left\||\mathbf{T}|_\sigma^{(d)} - \mathbf{Y}|_\sigma^{(d)}\right\|_\infty}{2^d}$$

$$\geq \frac{\left|(\mathbf{T}|_\sigma)_m^{(d)}\right| - \left|(\mathbf{Y}|_\sigma)_m^{(d)}\right|}{2^d}$$

$$\geq \frac{\left|(\mathbf{T}|_\sigma)_m^{(d)}\right|}{2^d} - \frac{n_\mathbb{R}\|\tilde{C}_\mathbb{R}^T \odot \tilde{B}_\mathbb{R}\|_\infty}{2^{d+2\min(m,d)}}.$$

Plugging in $||\mathbf{T}|_\sigma - \mathbf{Y}|_\sigma|_\infty \leq \epsilon$ and $\mathbf{T}|_\sigma$, we get

$$\epsilon \geq 2^{-d}\left|(\phi(\mathbf{I})|_\sigma)_m^{(d)}\right| - \frac{n_\mathbb{R}\|\tilde{C}_\mathbb{R}^T \odot \tilde{B}_\mathbb{R}\|_\infty}{2^{d+2\min(m,d)}}.$$

Reordering this equation yields the desired result. □

### D.4 Proof of Corollary 1

For simplicity of notation, we will denote $\left\lfloor\frac{t-1}{2}\right\rfloor$ by $k$ and assume without loss of generality that $k$ is odd. Theorem 2 indicates that

$$n_\mathbb{R}\|C_\mathbb{R}^\top \odot B_\mathbb{R}\|_\infty \geq \max_{d,m\in\mathbb{N}, d+m\leq\lfloor t/2\rfloor, \sigma\in\{\text{odd}, \text{even}\}} \left\{2^{d+2\min\{d,m\}}\left(2^{-d}\left|(\phi(\mathbf{I})|_\sigma)_m^{(d)}\right| - \epsilon\right)\right\}$$

Substituting $d = \frac{k+1}{2}$, $m = \left\lfloor\frac{k+1}{4}\right\rfloor$, $\sigma = $ even, and $\epsilon \leq \frac{1}{8\sqrt{t}}$ yields:

$$n\|C^T \odot B\|_\infty \geq 2^{k-1}\left(\frac{\left|(\phi(\mathbf{I})|_\text{even})_{\left\lfloor\frac{k+1}{4}\right\rfloor}^{\left(\frac{k+1}{2}\right)}\right|}{2^{\frac{k+1}{2}}} - \frac{1}{8\sqrt{t}}\right) \tag{8}$$

In the subsequent analysis, we will demonstrate that

$$\left|(\phi(\mathbf{I})|_\text{even})_{\left\lfloor\frac{k+1}{4}\right\rfloor}^{\left(\frac{k+1}{2}\right)}\right| \geq \frac{2^{\frac{k+1}{2}}}{4\sqrt{t}}.$$

Once established, the remainder of the proof will naturally follow since plugging this forthcoming result into Equation (8) will ensure that:

$$n\|C^T \odot B\|_\infty \geq \frac{2^{k-1}}{8\sqrt{t}} = \frac{2^k}{16\sqrt{t}} = \frac{2^{\lfloor(t-1)/2\rfloor}}{16\sqrt{t}} \geq \frac{2^{(t/2)-1}}{16\sqrt{t}} = \frac{2^{(t/2)}}{32\sqrt{t}}$$

satisfying the requirements for our conclusion. Indeed, we have that for any $j \leq k$

$$(\phi(\mathbf{I})|_\text{even})_j = (\phi(\mathbf{I}))_{2j} = \delta_k(\mathbf{I})_{2j} = \mathbb{1}(2j = k+1) = \mathbb{1}\left(j = \frac{k+1}{2}\right).$$

Lemma 7 implies that

$$(\phi(\mathbf{I})|_\text{even})_{\left\lfloor\frac{k+1}{4}\right\rfloor}^{\left(\frac{k+1}{2}\right)} = \sum_{j=0}^{\frac{k+1}{2}} (\phi(\mathbf{I})|_\text{even})_{\left\lfloor\frac{k+1}{4}\right\rfloor+j} (-1)^{\frac{k+1}{2}-j}\binom{\frac{k+1}{2}}{j}$$

$$= \sum_{j=0}^{\frac{k+1}{2}} \mathbb{1}\left(j + \left\lfloor\frac{k+1}{4}\right\rfloor = \frac{k+1}{2}\right)(-1)^{\frac{k+1}{2}-j}\binom{\frac{k+1}{2}}{j}$$

$$= (-1)^{\left\lfloor\frac{k+1}{4}\right\rfloor}\binom{\frac{k+1}{2}}{\frac{k+1}{2} - \left\lfloor\frac{k+1}{4}\right\rfloor}$$

$$= (-1)^{\left\lfloor\frac{k+1}{4}\right\rfloor}\binom{\frac{k+1}{2}}{\left\lfloor\frac{k+1}{4}\right\rfloor}.$$

By Lemma 11, we can assert:

$$\left|(\phi(\mathbf{I})|_{\text{even}})^{\left(\frac{k+1}{2}\right)}_{\left\lfloor\frac{k+1}{4}\right\rfloor}\right| \geq \frac{2^{\frac{k+1}{2}}}{4\sqrt{2^{\frac{k+1}{2}}}} \geq \frac{2^{\frac{k+1}{2}}}{4\sqrt{t}},$$

which completes the proof. □

### D.5 Proof of Corollary 2

Theorem 2 indicates that

$$n_{\mathbb{R}}\|C_{\mathbb{R}}^{\top} \odot B_{\mathbb{R}}\|_\infty \geq \max_{d,m\in\mathbb{N},\, d+m\leq\lfloor t/2\rfloor,\, \sigma\in\{\text{odd},\text{even}\}} \left\{2^{d+2\min\{d,m\}}\left(2^{-d}\left|(\mathbf{R}_{:t}|\sigma)^{(d)}_m\right| - \epsilon\right)\right\}.$$

For any $\sigma \in \{\text{odd},\text{even}\}$, plugging in $m = \left\lfloor\frac{t}{8}\right\rfloor$, $d = \left\lfloor\frac{t}{4}\right\rfloor$, and $\epsilon \leq \frac{\alpha\sqrt{\delta}}{\sqrt{t}}$ yields:

$$n\|C^T \odot B\|_\infty \geq 2^{\frac{t}{2}-3}\left(\frac{\left|(\mathbf{R}_{:t}|\sigma)^{\left(\left\lfloor\frac{t}{4}\right\rfloor\right)}_{\left\lfloor\frac{t}{8}\right\rfloor}\right|}{2^{\left\lfloor\frac{t}{4}\right\rfloor}} - \frac{\alpha\sqrt{\delta}}{\sqrt{t}}\right). \tag{9}$$

In the subsequent analysis, we will demonstrate that for any $\sigma \in \{\text{odd},\text{even}\}$, the following holds with a probability smaller than $\sqrt{\delta}$:

$$\left|(\mathbf{R}_{:t}|\sigma)^{\left(\left\lfloor\frac{t}{4}\right\rfloor\right)}_{\left\lfloor\frac{t}{8}\right\rfloor}\right| \leq 2^{\left\lfloor\frac{t}{4}\right\rfloor} \cdot \frac{2\alpha\sqrt{\delta}}{\sqrt{t}}.$$

Once established, the remainder of the proof will naturally follow, as plugging this forthcoming result into Equation (9) will ensure that for any $\sigma \in \{\text{odd},\text{even}\}$, with probability $1 - \sqrt{\delta}$ we have that:

$$n\|C^T \odot B\|_\infty \geq \frac{2^{\frac{t}{2}-3}\alpha\sqrt{\delta}}{\sqrt{t}} = \frac{2^{\frac{t}{2}}\alpha\sqrt{\delta}}{8\sqrt{t}},$$

satisfying the requirements for our conclusion. Since $\mathbf{R}_{:t}|_{\text{even}}$ and $\mathbf{R}_{:t}|_{\text{odd}}$ are independent random variables, we have that the previous statement holds with probability higher than $1 - \delta$ as required.

Now let us proceed with proving the aforementioned result. Assume without loss of generality that $\sigma = \text{even}$. Lemma 7 implies that

$$\left|(\mathbf{R}|_{\text{even}})^{\left(\left\lfloor\frac{t}{4}\right\rfloor\right)}_{\left\lfloor\frac{t}{8}\right\rfloor}\right| = \left|\sum_{j=0}^{\left\lfloor\frac{t}{4}\right\rfloor}(\mathbf{R}|_{\text{even}})_{\left\lfloor\frac{t}{8}\right\rfloor+j}(-1)^{\left\lfloor\frac{t}{4}\right\rfloor-j}\binom{\left\lfloor\frac{t}{4}\right\rfloor}{j}\right|$$

$$= \left|\sum_{j=0}^{\left\lfloor\frac{t}{4}\right\rfloor}(\mathbf{R})_{2\left\lfloor\frac{t}{8}\right\rfloor+2j}(-1)^{\left\lfloor\frac{t}{4}\right\rfloor-j}\binom{\left\lfloor\frac{t}{4}\right\rfloor}{j}\right|.$$

For each $j \in \left[\left\lfloor\frac{t}{4}\right\rfloor\right]$, define the random variable $X_j$ as follows:

$$X_j = (\mathbf{R})_{2\left\lfloor\frac{t}{8}\right\rfloor+2j}(-1)^{\left\lfloor\frac{t}{4}\right\rfloor-j}\binom{\left\lfloor\frac{t}{4}\right\rfloor}{j},$$

and denote its PDF by $p_{X_j}$. We have that

$$X_j \sim \binom{\left\lfloor\frac{t}{4}\right\rfloor}{j}\mathcal{U}(-\alpha,\alpha) = \mathcal{U}\left(-\alpha\binom{\left\lfloor\frac{t}{4}\right\rfloor}{j},\alpha\binom{\left\lfloor\frac{t}{4}\right\rfloor}{j}\right),$$

which means that

$$p_{X_j}(x) = \begin{cases} \frac{1}{2\alpha\binom{\left\lfloor\frac{t}{4}\right\rfloor}{j}} & \text{if } -\alpha \leq x \leq \alpha \\ 0 & \text{otherwise} \end{cases}.$$

Let $X$ be the sum of these random variables $X = \sum_{j=0}^{\lfloor \frac{t}{4} \rfloor} X_j$ and denote its PDF by $p_X$. We note that

$$P\left[ \left| (\mathbf{R}|_{\text{even}})_{\lfloor \frac{t}{8} \rfloor}^{(\lfloor \frac{t}{4} \rfloor)} \right| \le \epsilon_0 \right] = P[-\epsilon_0 < X < \epsilon_0] \le 2\epsilon_0 \max_{x \in \mathbb{R}} p_X(x).$$

As a sum of i.i.d. random variables, the maximum of $p_X$ is upper bounded by the maximum of the PDF of each of the random variables it is a sum of. Which means that

$$\max_{x \in \mathbb{R}} p_X(x) \le \max_{x \in \mathbb{R}} p_{X_{\lfloor \frac{t}{8} \rfloor}}(x) = \frac{1}{2\alpha \binom{\lfloor t/4 \rfloor}{\lfloor t/8 \rfloor}} \le \frac{\sqrt{\lfloor \frac{t}{8} \rfloor}}{\alpha 2^{\lfloor \frac{t}{4} \rfloor}} \le \frac{\sqrt{t}}{2\alpha 2^{\lfloor \frac{t}{4} \rfloor}}$$

(where the second inequality results from Lemma 10). Overall we have that

$$P\left[ \left| (\mathbf{R}|_{\text{even}})_{\lfloor \frac{t}{8} \rfloor}^{(\lfloor \frac{t}{4} \rfloor)} \right| \le \epsilon_0 \right] \le \frac{\epsilon_0 \sqrt{t}}{2\alpha 2^{\lfloor \frac{t}{4} \rfloor}}.$$

Plugging in $\epsilon_0 = 2^{\lfloor \frac{t}{4} \rfloor} \cdot \frac{2\alpha\sqrt{\delta}}{\sqrt{t}}$ we have that

$$P\left[ \left| (\mathbf{R}|_{\text{even}})_{\lfloor \frac{t}{8} \rfloor}^{(\lfloor \frac{t}{4} \rfloor)} \right| \le 2^{\lfloor \frac{t}{4} \rfloor} \cdot \frac{2\alpha\sqrt{\delta}}{\sqrt{t}} \right] \le \sqrt{\delta}$$

which concludes the proof. $\square$

### D.6 Proof of Corollary 3

Theorem 2 demonstrated that

$$n_{\mathbb{R}} \| C_{\mathbb{R}}^\top \odot B_{\mathbb{R}} \|_\infty \ge \max_{d,m \in \mathbb{N},\, d+m \le \lfloor t/2 \rfloor,\, \sigma \in \{\text{odd},\text{even}\}} \left\{ 2^{d+2\min\{d,m\}} \left( 2^{-d} \left| (\phi(\mathbf{I})|_\sigma)_m^{(d)} \right| - \epsilon \right) \right\}.$$

Plugging in $\sigma = \text{odd}$, we have that for any $m \in [\lfloor t/2 \rfloor]$ the following holds:

$$(\phi(\mathbf{I})|_\sigma)_m = (-1)^{m-1}.$$

Using simple induction, it is easy to show that for any $d, m \in \mathbb{N}$ such that $d + m \le \lfloor t/2 \rfloor$,

$$(\phi(\mathbf{I})|_\sigma)_m^{(d)} = (-1)^{(m+d-1)} 2^d.$$

Plugging this into Theorem 2 with $\sigma = \text{odd}$, $d = \lfloor t/4 \rfloor$, $m = \lfloor t/4 \rfloor$, and $\epsilon = 0.5$ yields:

$$n_{\mathbb{R}} \| C_{\mathbb{R}}^\top \odot B_{\mathbb{R}} \|_\infty \ge 2^{\lfloor t/4 \rfloor + 2\lfloor t/4 \rfloor} (1 - 0.5) = 2^{3\lfloor t/4 \rfloor - 1} \ge 2^{3t/4 - 4},$$

as required.

$\square$

### D.7 Proof of Proposition 3

It is sufficient to prove the proposition for $n_{\mathbb{C}} = t$ since the dimension of a diagonal SSM can always be effectively reduced by zeroing out elements of $B_{\mathbb{C}}$.

According to Section 2.3, it is enough to show that there exist assignments for $(A_{\mathbb{C}}, B_{\mathbb{C}}, C_{\mathbb{C}})$ such that the following conditions hold:

$$\phi_{t,(A_{\mathbb{C}}, B_{\mathbb{C}}, C_{\mathbb{C}})}(\mathbf{I})_{:t} = \phi(\mathbf{I})_{:t}, \quad \|B_{\mathbb{C}}\|_2 = 2\|\phi(\mathbf{I})_{:t}\|_2, \quad \|C_{\mathbb{C}}^T\|_2 = 1.$$

For convenience, we will denote $A_{\mathbb{C}}, B_{\mathbb{C}}, C_{\mathbb{C}}$ by $A, B, C$.

We begin by utilizing the theory of *discrete Fourier transform* (*DFT*) to allow us to write the truncated impulse response of $\phi$ of length $t$ as a linear combination of $t$ decaying sine and cosine waves. Specifically, defining $\left( \tilde{\phi}(\mathbf{I})_{:t} \right)_k = \frac{(\phi(\mathbf{I})_{:t})_k}{\alpha^{k-1}}$, where $\alpha \in (0, 1)$ is some constant, and denoting the DFT of $\tilde{\phi}(\mathbf{I})_{:t}$ by $a + bi \in \mathbb{C}^t$ we get that

$$\forall k \in [t], \frac{(\phi(\mathbf{I})_{:t})_k}{\alpha^{k-1}} = \frac{1}{t} \sum_{j=0}^{t-1} a_{j+1} \cos\left( 2\pi(k-1)\frac{j}{t} \right) + \sum_{j=0}^{t-1} b_{j+1} \sin\left( 2\pi(k-1)\frac{j}{t} \right).$$

Now, we will derive assignments for the SSM's parameters so that its impulse response will equate this sum. Indeed, we fix the diagonal entries of $A$ to be a scaled version of the t roots of unity, and $C$ to be the inverse of the square root of $t$ as follows:

$$A_{j,j} = \alpha e^{2\pi i \frac{j-1}{t}}, \quad C_j = \frac{1}{\sqrt{t}}$$

We know (see Section 2.3) that the impulse response of a complex system at index $k$ is given by:

$$\left(\phi_{t,(A,B,C)}(\mathbf{I})_{:t}\right)_k = \Re\left(\sum_{j=1}^{t} A_{j,j}^{k-1} B_j C_j\right).$$

Which implies that

$$\Re\left(\sum_{j=1}^{t} A_{j,j}^{k-1} B_j C_j\right) = \sum_{j=1}^{t} \Re(C_j A_{j,j}^{k-1})\Re(B_j) - \Im(C_j A_{j,j}^{k-1})\Im(B_j)$$

$$= \alpha^{k-1} \frac{1}{\sqrt{t}} \left(\sum_{j=0}^{t-1} \cos\left(2\pi(k-1)\frac{j}{t}\right)\Re(B_j) - \sum_{j=0}^{t-1} \sin\left(2\pi(k-1)\frac{j}{t}\right)\Im(B_j)\right).$$

By the Plancherel theorem the following holds:

$$\|a+bi\|_2 = \sqrt{t}\left\|\tilde{\phi}(\mathbf{I})_{:t}\right\|_2 \leq \frac{\sqrt{t}\|\phi(\mathbf{I})_{:t}\|_2}{\alpha^{t-1}}.$$

Therefore, if we define $B = \frac{a-ib}{\sqrt{t}}$, we get

$$\Re\left(\sum_{j=1}^{t} A_{j,j}^{k-1} B_j C_j\right) = \frac{\alpha^{k-1}}{t} \left(\sum_{j=0}^{t-1} a_{j+1} \cos\left(2\pi(k-1)\frac{j}{t}\right) + \sum_{j=0}^{t-1} b_{j+1} \sin\left(2\pi(k-1)\frac{j}{t}\right)\right)$$

$$= (\phi(\mathbf{I})_{:t})_k$$

and

$$\|B\|_2 = \left\|\frac{a-bi}{\sqrt{t}}\right\|_2 \leq \frac{\|\phi(\mathbf{I})_{:t}\|_2}{\alpha^{t-1}}, \quad \|C\|_2 = \sqrt{t \cdot \left(\frac{1}{\sqrt{t}}\right)^2} = 1,$$

and by choosing $\alpha = \left(\frac{1}{2}\right)^{\frac{1}{t-1}}$ we get the required assignment. $\qquad\square$

### D.8 Proof of Proposition 4

For simplicity of notation, let $A_{\mathbb{R}}^{(i)}, B_{\mathbb{R}}^{(i)}, C_{\mathbb{R}}^{(i)}$ be denoted by $A^{(i)}, B^{(i)}, C^{(i)}$, respectively, and $\ell(A^{(i)}, B^{(i)}, C^{(i)})$ by $\ell^{(i)}$. We begin by writing the loss explicitly (see Appendix C):

$$\ell(A, B, C) = \sum_{j=1}^{t} \left(\sum_{k=1}^{n_\mathbb{R}} A_k^{j-1} B_k C_k - \phi(\mathbf{I})_j\right)^2.$$

Fixing any $k \in [n_\mathbb{R}]$, we have

$$\frac{d\ell(A, B, C)}{dB_k} = 2\sum_{j=1}^{t} A_k^{j-1} C_k \left(\sum_{m=1}^{n_\mathbb{R}} A_m^{j-1} B_m C_m - \phi(\mathbf{I})_j\right),$$

which, combined with the fact that $|A_m| \leq 1$, leads to

$$\left|\frac{d\ell(A, B, C)}{dB_k}\right| \leq 2|C_k| \sum_{j=1}^{t} \left|\sum_{m=1}^{n_\mathbb{R}} A_m^{j-1} B_m C_m - \phi(\mathbf{I})_j\right|$$

$$= 2|C_k| \left\|\phi_{n,(A,B,C)}(\mathbf{I})_{:t} - \phi(\mathbf{I})_{:t}\right\|_1$$

$$\leq 2\sqrt{t}|C_k| \left\|\phi_{n,(A,B,C)}(\mathbf{I})_{:t} - \phi(\mathbf{I})_{:t}\right\|_2$$

$$\leq 2\sqrt{t} \cdot \max\left(\|B\|_\infty, \|C\|_\infty\right) \sqrt{\ell(A, B, C)}.$$

This implies

$$\|B^{(i)}\|_\infty \leq \|B^{(i-1)}\|_\infty + \eta^{(i)} \cdot 2\sqrt{t \cdot \ell^{(i-1)}} \max\left(\left\|B^{(i-1)}\right\|_\infty, \left\|C^{(i-1)}\right\|_\infty\right).$$

Next, we will demonstrate that

$$\eta^{(i)} \max\left(\left\|B^{(i-1)}\right\|_\infty, \left\|C^{(i-1)}\right\|_\infty\right) \leq \sqrt{c_1},$$

which concludes the proof since this aforementioned result will establish that for all $i \in \mathbb{N}$, the following holds:

$$
\begin{aligned}
\|B^{(i)}\|_\infty &\leq \|B^{(i-1)}\|_\infty + 2\sqrt{t\ell^{(i-1)}c_1} \\
&\leq \|B^{(i-1)}\|_\infty + 2\sqrt{tc_2\ell^{(0)}c_1} \tag{1} \\
&\leq \|B^{(0)}\|_\infty + 2i\sqrt{tc_2\ell^{(0)}c_1}. \tag{2}
\end{aligned}
$$

Where (1) holds because we assume $\ell^{(i-1)} \leq c_2\ell^{(0)}$ and (2) is by recursion.

Repeating the same argument for $C^{(i)}$ completes the proof.

Finally, let us show that $\eta^{(i)} \max\left(\left\|B^{(i-1)}\right\|_\infty, \left\|C^{(i-1)}\right\|_\infty\right) \leq \sqrt{c_1}$. Indeed, Lemma 3 combined with the assumptions that

$$\forall i \in \mathbb{N} : \eta^{(i)} \leq \frac{2}{\max\left\{\lambda_{max}\nabla^2\ell^{(i-1)}, 0\right\}}$$

and $\forall i \in \mathbb{N} : \eta^{(i)} \leq c_1$ gives that

$$\eta^{(i)} \leq \min\left\{\frac{2}{2\max\left\{\left\|B^{(i-1)}\right\|_\infty, \left\|C^{(i-1)}\right\|_\infty\right\}^2}, c_1\right\}.$$

Considering the cases where $\max\left\{\left\|B^{(i-1)}\right\|_\infty, \left\|C^{(i-1)}\right\|_\infty\right\}$ is bigger and smaller than $1/\sqrt{c_1}$ we immediately get the required result. $\qquad\square$

**Lemma 3.** *In the setting of Proposition 4 we have that for any $A_\mathbb{R}, B_\mathbb{R}, C_\mathbb{R}$ the following holds:*

$$\lambda_{max}\left(\nabla^2\ell\left(A_\mathbb{R}, B_\mathbb{R}, C_\mathbb{R}\right)\right) \geq 2\max\left\{\|B_\mathbb{R}\|_\infty, \|C_\mathbb{R}\|_\infty\right\}^2$$

*Proof.* For simplicity of notation, let $A_\mathbb{R}, B_\mathbb{R}, C_\mathbb{R}$ be denoted by $A, B, C$, respectively. We will assume, without loss of generality, that $\|C\|_\infty \geq \|B\|_\infty$, and denote $i = \arg\max_i C_i$ and by $e_i$ the one-hot vector with 1 at index $i$ and zero elsewhere. We will also denote the following function

$$I_i(x) = \phi_{n_\mathbb{R}, (A, B + x \cdot e_i, C)}(\mathbf{I})_{:t}$$

by $I_i(x)$, and the following loss function:

$$\tilde{\ell}_i(x) = \ell(I_i(x)) = \tilde{\ell}(A, B + x \cdot e_i, C)$$

by $\tilde{\ell}_i$. Next, We notice that by definition of the Hessian, we have that

$$\left(\nabla^2\ell(A, \cdot, C)[B]\right)_{i,i} = \tilde{\ell}_i''(0).$$

Therefore, $\lambda_{max}\left(\nabla^2\ell\right)$ at $A, B, C$ is lower-bounded by $\tilde{\ell}_i''(0)$. It is therefore sufficient to show that $|\tilde{\ell}_i''(0)| \geq 2\|C\|_\infty^2$. Indeed, by Appendix C, the following holds

$$|\tilde{\ell}_i''(0)| = \left| \frac{d\ell(A, B, C)}{(dB_i)^2} \right|$$

$$= \left| \frac{d\left( \sum_{j=1}^t \left( \sum_{m=1}^{n_{\mathbb{R}}} A_m^{j-1} B_m C_m - \phi(\mathbf{I})_j \right)^2 \right)}{(dB_i)^2} \right|$$

$$= \left| \sum_{j=1}^t \frac{d}{dB_i} \left( \left( \sum_{m=1}^{n_{\mathbb{R}}} A_m^{j-1} B_m C_m - \phi(\mathbf{I})_j \right) \cdot 2A_i^{j-1} C_i \right) \right|$$

$$= \left| \sum_{j=1}^t 2A_i^{j-1} C_i \cdot A_i^{j-1} C_i \right|$$

$$\geq 2\|C\|_\infty^2 \,,$$

which concludes the proof. $\qquad\square$

### D.9 Proof of Proposition 5

For simplicity of notation, we will denote $A_{\mathbb{R}}, B_{\mathbb{R}}, C_{\mathbb{R}}, n_{\mathbb{R}}$ by $A, B, C, n$. The robustness to $q$-quantization is by definition the probability:

$$\mathrm{P}\left[ \|\phi_{n,(A \odot Q_A, B \odot Q_B, C \odot Q_C)}(\mathbf{I})_{:t} - \phi(\mathbf{I})_{:t}\|_1 \leq \epsilon \right]$$
$$\leq \mathrm{P}\left[ \left| \phi_{n,(A \odot Q_A, B \odot Q_B, C \odot Q_C)}(\mathbf{I})_1 - \phi(\mathbf{I})_1 \right| \leq \epsilon \right] .$$

Let us denote the random variable $\phi_{n,(A \odot Q_A, B \odot Q_B, C \odot Q_C)}(\mathbf{I})_1$ by $X$, and its PDF by $p_X$. The probability $\mathrm{P}\left[ |X - Y| \leq \epsilon \right]$ is equal to the integral $\int_{Y-\epsilon}^{Y+\epsilon} p_X(x)dx \leq 2\epsilon \max_{x \in \mathbb{R}} p_X(x)$. We get that

$$\mathrm{P}\left[ \|\phi_{n,(A \odot Q_A, B \odot Q_B, C \odot Q_C)}(\mathbf{I}) - \phi(\mathbf{I})\|_1 \leq \epsilon \right] \leq 2\epsilon \max_{x \in \mathbb{R}} p_X(x) \,.$$

and so, to prove the theorem, it suffices to show that

$$\max_{x \in \mathbb{R}} p_X(x) \leq \frac{1}{q\|B \odot C\|_\infty} \,.$$

Let us investigate the random variable $X$:

$$X = \sum_{i=1}^n (B \odot Q_B)_i (C \odot Q_C)_i = \sum_{i=1}^n B_i C_i(1 + q_{B_i})(1 + q_{C_i})$$
$$= \sum_{i=1}^n B_i C_i(1 + q_{B_i} + q_{C_i} + q_{B_i} q_{C_i}) \,,$$

where $q_{B_i}, q_{C_i} \sim \mathcal{U}(-q/2, q/2)$. As a sum of independent random variables, the maximum of $p_X$ is upper bounded by the maximum of the PDF of each of the random variables it is a sum of. Specifically, for any $i \in [n]$, it is bounded by the maximum of the PDF of $\mathcal{U}(-B_i C_i q/2, B_i C_i q/2)$ which is equal to $1/(B_i C_i q)$. Overall, this means that

$$\max_{x \in \mathbb{R}} p_X(x) \leq \min_i \frac{1}{B_i C_i q} = \frac{1}{q\|B \odot C\|_\infty} \,,$$

as needed. $\qquad\square$

# E  Auxiliary Lemmas

**Lemma 4.**  *For any $n$ non-negative numbers $a_1, \ldots, a_n$, and $n$ real numbers $b_1, \ldots, b_n$, the function $f$ defined below*

$$f(x) = \sum_{i=1}^{n} b_i a_i{}^x$$

*can have at most $n-1$ zeros.*

*Proof.* We will show this by induction. The claim is obvious for $n = 1$, let us assume the induction claim is true for $n = k$ and we will show that it is true for $n = k + 1$. Indeed, let $f$ be the function defined by

$$f(x) = \sum_{i=1}^{k+1} b_i a_i{}^x.$$

If $b_1 = 0$ or $a_1 = 0$ we can immediately use the induction step on $f$. Otherwise, we can write $f$ in the following way:

$$f(x) = b_1 a_1^x \left( 1 + \sum_{i=2}^{k+1} \frac{b_i}{b_1} \left( \frac{a_i}{a_1} \right)^x \right).$$

Since $b_1 a_1^x$ is always non-zero, the amount of roots of $f$ is equal to the number of roots of the function $g$ defined by

$$g(x) = 1 + \sum_{i=2}^{n+1} \frac{b_i}{b_1} \left( \frac{a_i}{a_1} \right)^x.$$

By Rolle's theorem, the number of roots of a function is bounded by the number of roots of its derivative plus one. We have that $g'(x)$ is given by:

$$g'(x) = \sum_{i=2}^{k+1} \frac{b_i}{b_1} \left( \frac{a_i}{a_1} \right)^x \ln \left( \frac{a_i}{a_1} \right).$$

Which, by the induction step, can have at most $k$ roots, as needed. $\square$

**Lemma 5.**  *Let $\theta$ be a real number in the open interval $(0, 2\pi)$, let $b$ be a real number in $[0, 2\pi]$, and let $p$ be a natural number such that $\theta \cdot p \geq 2\pi$. Then the set*

$$\{(b + \theta) \bmod 2\pi, (b + 2\theta) \bmod 2\pi, \ldots, (b + p \cdot \theta) \bmod 2\pi\}$$

*contains at least one point in any closed and continuous-mod-$2\pi$ interval of length $\theta$ within $[0, 2\pi]$.*

*Proof.* Let $x_k = (b + k\theta)$ for $k = 1, 2, \ldots, p$. Define the intervals

$$I_k = [x_{k-1}, x_k) \bmod 2\pi \quad \text{for} \quad k = 1, 2, \ldots, p.$$

Each interval $I_k$ has a length of $\theta$. Since $p\theta \geq 2\pi$ we have that

$$\bigcup_{k=1}^{p} I_k = [0, 2\pi].$$

Now, consider any closed and continuous-mod-$2\pi$ interval $J \subset [0, 2\pi)$ of length $\theta$. Since $\bigcup_{k=1}^{p} I_k = [0, 2\pi]$, the interval $J$ must intersect at least one of the intervals $I_k$. That is,

$$J \cap I_{k^*} \neq \emptyset \quad \text{for some} \quad k^* \in \{1, 2, \ldots, p\}.$$

Since $J$ has length $\theta$ and overlaps with $I_{k^*}$ (which also has length $\theta$), and since both are continuous-mod-$2\pi$ their intersection must include at least one of the endpoints of $I_{k^*}$ which is in the set

$$\{(b + \theta) \bmod 2\pi, (b + 2\theta) \bmod 2\pi, \ldots, (b + p \cdot \theta) \bmod 2\pi\}.$$

This completes the proof. $\square$

**Lemma 6.** *Let $\alpha$ be a real number within the interval $[0,1]$, and let $n, m$ be natural numbers with $t$ being a natural number such that $t \geq n + m$. The absolute value of the $m$th forward difference of the sequence $X = \{1, \alpha, \alpha^2, \ldots, \alpha^{t-1}\}$ at index $n$, as defined in Definition 1, is bounded above by*

$$\left(\frac{m}{n+m}\right)^m \left(\frac{n}{n+m}\right)^n.$$

*Proof.* By induction, the $m$th forward difference of $X$ at index $n$ (when we start at index 0), for $n \leq t - m$, is $\alpha^n(\alpha - 1)^m$. For $\alpha \in [0,1]$, this absolute value simplifies to:

$$|\alpha^n(\alpha - 1)^m| = \alpha^n(1 - \alpha)^m.$$

To find the maximum value of this expression within $[0,1]$, we differentiate with respect to $\alpha$ and set the derivative to zero:

$$\frac{d}{d\alpha}(\alpha^n(1 - \alpha)^m) = n\alpha^{n-1}(1 - \alpha)^m - m\alpha^n(1 - \alpha)^{m-1}.$$

Solving for $\alpha$, we find that the critical points are at $\alpha \in \{0, 1\}$ or:

$$n(1 - \alpha) - m\alpha = 0 \implies \alpha = \frac{n}{n+m}.$$

A second derivative test confirms that $\alpha = \frac{n}{n+m}$ is a maximum within $[0,1]$. Substituting $\alpha_{\max} = \frac{n}{n+m}$ into our original expression, we obtain:

$$|(X^{(m)})_n| \leq \left(\frac{n}{n+m}\right)^n \left(1 - \frac{n}{n+m}\right)^m = \left(\frac{m}{n+m}\right)^m \left(\frac{n}{n+m}\right)^n,$$

thereby completing the proof. $\square$

**Lemma 7.** *Given a sequence $A = \{a_0, a_1, \ldots, a_{n-1}\}$ and two natural numbers $m < n$, the $m$th forward difference of $A$ (Definition 1) at index $n$, denoted $\left(A^{(m)}\right)_n$, is given by*

$$\left(A^{(m)}\right)_n = \sum_{j=0}^m a_{n+j}(-1)^{m-j}\binom{m}{j}$$

*Proof.* We will show this by induction over $m$. For $m = 0$ This is obvious. Let's assume that the Lemma is true for $m = k$ and prove that it is also true for $m = k + 1 < n$.

Indeed

$$\left(A^{(m+1)}\right)_n = \left(A^{(m)}\right)_{n+1} - \left(A^{(m)}\right)_n$$

$$= \sum_{j=0}^m (-1)^{m-j} a_{n+j+1}\binom{m}{j} - \sum_{j=0}^m (-1)^{m-j} a_{n+j}\binom{m}{j}$$

$$= a_{n+m+1} - (-1)^m a_n + \sum_{j=0}^{m-1} (-1)^{m-j} a_{n+j+1}\left(\binom{m}{j} + \binom{m}{j+1}\right)$$

$$= a_{n+m+1} - (-1)^m a_n + \sum_{j=0}^{m-1} (-1)^{m-j} a_{n+j+1}\binom{m+1}{j+1}$$

$$= a_{n+m+1} - (-1)^m a_n + \sum_{j=1}^m (-1)^{m-j+1} a_{n+j}\binom{m+1}{j}$$

$$= \sum_{j=0}^{m+1} (-1)^{m+1-j} a_{n+j}\binom{m+1}{j}$$

as needed. $\square$

**Lemma 8.** *Given two sequences $A = \{a_0, a_1, \ldots, a_{n-1}\}$ and $B = \{b_0, b_1, \ldots, b_{n-1}\}$ of length $n$, and for any natural number $m < n$, it holds that*

$$\|A - B\|_\infty \geq \frac{\|A^{(m)} - B^{(m)}\|_\infty}{2^m}$$

*where $A^{(m)}$ and $A^{(m)}$ are the $m$th forward difference (Definition 1) of $A$ and $B$ respectively*

*Proof.* We aim to demonstrate that

$$2^m \|A - B\|_\infty \geq \|A^{(m)} - B^{(m)}\|_\infty$$

which directly supports the statement of the lemma.

It suffices to verify this for $m = 1$, as the general case can then be established by induction, showing that:

$$\|A^{(m)} - B^{(m)}\|_\infty \leq 2\|A^{(m-1)} - B^{(m-1)}\|_\infty \leq \cdots \leq 2^m \|A - B\|_\infty$$

Indeed, for $m = 1$, we have that for any $i < n - 1$

$$|(A^{(1)} - B^{(1)})_i| = |a_{i+1} - a_i - b_{i+1} + b_i| \leq |a_{i+1} - b_{i+1}| + |a_i - b_i| \leq 2\|A - B\|_\infty$$

$\square$

**Lemma 9.** *Let $\phi_{n_\mathbb{R}, (A_\mathbb{R}, B_\mathbb{R}, C_\mathbb{R})}(\mathbf{I})_{:t}$ be a truncated impulse response of a real SSM denoted by $\mathbf{Y}$. Each of the following two sequences*

$$\mathbf{Y}|_{odd} = (\mathbf{Y}_1, \mathbf{Y}_3, \ldots)$$

*and*

$$\mathbf{Y}|_{even} = (\mathbf{Y}_2, \mathbf{Y}_4, \ldots)$$

*are impluse responses of some other real SSM of dimension $n_\mathbb{R}$ with parameters $(\tilde{A}_\mathbb{R}, \tilde{B}_\mathbb{R}, \tilde{C}_\mathbb{R})$ where $\tilde{A}_\mathbb{R}$ is non-negative, and*

$$\|C_\mathbb{R}^T \odot B_\mathbb{R}\|_\infty \geq \|\tilde{C}_\mathbb{R}^T \odot \tilde{B}_\mathbb{R}\|_\infty$$

*Proof.* We observe that (see Section 2.3)

$$(\mathbf{Y}|_{odd})_i = C_\mathbb{R} A_\mathbb{R}^{2(i-1)} B_\mathbb{R}$$

setting $(\tilde{A}_\mathbb{R}, \tilde{B}_\mathbb{R}, \tilde{C}_\mathbb{R}) = (A_\mathbb{R}^2, B_\mathbb{R}, C_\mathbb{R})$ yields that $\mathbf{Y}|_{odd}$ is the impulse response of the real SSM $\phi_{n_\mathbb{R}, (\tilde{A}_\mathbb{R}, \tilde{B}_\mathbb{R}, \tilde{C}_\mathbb{R})}$.

For the even case:

$$(\mathbf{Y}|_{even})_i = C_\mathbb{R} A_\mathbb{R}^{2i-1} B_\mathbb{R}$$

setting $(\tilde{A}_\mathbb{R}, \tilde{B}_\mathbb{R}, \tilde{C}_\mathbb{R}) = (A_\mathbb{R}^2, B_\mathbb{R}, C_\mathbb{R} A_\mathbb{R})$ yields that $\mathbf{Y}|_{even}$ is the impulse response of the real SSM $\phi_{n_\mathbb{R}, (\tilde{A}_\mathbb{R}, \tilde{B}_\mathbb{R}, \tilde{C}_\mathbb{R})}$. In both cases, $\tilde{A}_\mathbb{R}$ is non-negative, and

$$\|C_\mathbb{R}^T \odot B_\mathbb{R}\|_\infty \geq \|\tilde{C}_\mathbb{R}^T \odot \tilde{B}_\mathbb{R}\|_\infty$$

since in the even case $(\tilde{B}_\mathbb{R}, \tilde{C}_\mathbb{R}) = (B_\mathbb{R}, C_\mathbb{R})$ and in the odd case $|\tilde{C}_{\mathbb{R}j}| \leq |C_{\mathbb{R}j}|$ because $|A_{\mathbb{R}j,j}| \leq 1$ for all $j$. $\square$

**Lemma 10.** *For any positive integer $n$, the following inequality holds:*

$$\binom{2n}{n} \geq \frac{2^{2n}}{2\sqrt{n}}$$

*Proof.* Using the Stirling's formula for factorials, we know that $n!$ is bounded by:

$$\sqrt{2\pi n}\left(\frac{n}{e}\right)^n e^{\left(\frac{1}{12n}-\frac{1}{360n^3}\right)} < n! < \sqrt{2\pi n}\left(\frac{n}{e}\right)^n e^{\frac{1}{12n}}.$$

which can be simplified to:

$$\sqrt{2\pi n}\left(\frac{n}{e}\right)^n < n! < \sqrt{2\pi n}\left(\frac{n}{e}\right)^n e^{\frac{1}{12n}}.$$

We can plug the bound in the factorial form of the binomial coefficients:

$$\binom{2n}{n} = \frac{(2n)!}{n!\cdot n!} > \frac{\sqrt{2\pi 2n}\left(\frac{2n}{e}\right)^{2n}}{(\sqrt{2\pi n})^2\left(\frac{n}{e}\right)^{2n} e^{\frac{1}{6n}}} = \frac{2^{2n}}{\sqrt{\pi n}e^{\frac{1}{6n}}} > \frac{2^{2n}}{2\sqrt{n}}$$

Where the last inequality is due to $\sqrt{\pi}e^{\frac{1}{6n}} < 2$ for all $n \geq 2$. To complete the proof for all positive integers notice we get equality for $n = 1$. $\square$

**Lemma 11.** *For any positive integer $n$, the following inequalitie hold:*

$$\binom{n}{\lfloor \frac{n}{2}\rfloor} \geq \frac{2^n}{4\sqrt{2n}}$$

*Proof.* Utilizing Lemma 10

we have

$$\binom{2m}{m} \geq \frac{2^{2m}}{2\sqrt{m}}.$$

To validate the lemma, we consider cases based on the parity of $n$. For even $n$,

$$\binom{n}{\lfloor \frac{n}{2}\rfloor} = \binom{n}{\frac{n}{2}} \geq \frac{2^n}{2\sqrt{\frac{n}{2}}} > \frac{2^n}{4\sqrt{2n}},$$

and for odd $n$,

$$\binom{n}{\lfloor \frac{n}{2}\rfloor} = \binom{n}{\frac{n-1}{2}} \geq \binom{n-1}{\frac{n-1}{2}} \geq \frac{2^{n-2}}{2\sqrt{\frac{n-1}{2}}} \geq \frac{2^n}{4\sqrt{2n}},$$

thereby proving the lemma as required. $\square$

**Lemma 12.** *For any positive integers $n$ and $m$, it holds that*

$$\left(\frac{m}{n+m}\right)^m\left(\frac{n}{n+m}\right)^n \leq \frac{1}{2^{2\min(m,n)}}$$

*Proof.* We establish this inequality through direct analysis:

$$\left(\frac{m}{n+m}\right)^m\left(\frac{n}{n+m}\right)^n \leq \left(\frac{m}{n+m}\right)^{\min(m,n)}\left(\frac{n}{n+m}\right)^{\min(m,n)}$$

$$= \left(\frac{mn}{(n+m)^2}\right)^{\min(m,n)}$$

$$\leq \left(\frac{mn}{4nm}\right)^{\min(m,n)}$$

$$= \frac{1}{2^{2\min(m,n)}}$$

This last inequality leverages the fact that $(n+m)^2 - 4nm = (n-m)^2 > 0$. $\square$

**Lemma 13.** *For any complex number $x = a + bi$ with $a \leq 0$, it holds that*

$$\left|\frac{1}{1-x}\right| \leq 1.$$

*Proof.* It suffices to show that $|1 - x| \geq 1$. Indeed, we have

$$|1 - x| = \sqrt{(1-a)^2 + b^2} \geq \sqrt{(1-a)^2} = |1 - a| = 1 - a \geq 1,$$

as required. $\square$

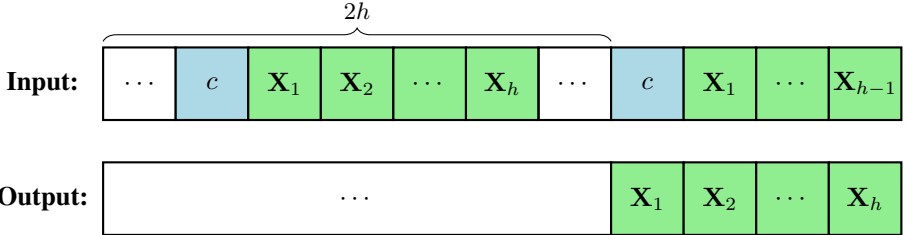

Figure 1: Illustration of the induction-head task. See Appendix F.3 for details.

# F  Implementation Details

Appendices F.1 to F.3 below present implementation details omitted from Sections 4.1 to 4.3, respectively. Source code for reproducing the results of Sections 4.1 and 4.3 can be found at `https://github.com/edenlum/SSMComplexParamBenefits`. The results of Section 4.2 were obtained using the official S4 repository, available at `https://github.com/state-spaces/s4` (Apache-2.0 license). All experiments were conducted on a single NVIDIA A6000 GPU.

## F.1  Theoretically Analyzed Setting

In all experiments, we parameterized state transition matrices in a manner that ensures stability, similarly to the LRU architecture [41]. For real SSMs, we performed a grid search for each optimizer, varying learning rates and initialization schemes. Namely, we evaluated learning rates of $1 \times 10^{-4}, 1 \times 10^{-5}$ and $1 \times 10^{-6}$, and randomly initialized the diagonal elements of $A_\mathbb{R}$ uniformly in $[-1, 1]$ or in $[-1, 0.99] \cup [0.99, 1]$. For complex SSMs, we used a learning rate of $1 \times 10^{-5}$ and initialized the diagonal elements of $A_\mathbb{C}$ similarly to [41], by sampling uniformly from the complex ring with radii 0.99 to 1. For all SSMs, we employed a cosine learning rate scheduler [35] and trained for half a million steps.

## F.2  Real-World Setting

Our implementation is based on the official S4 repository, available at `https://github.com/state-spaces/s4` (Apache-2.0 license). Unless stated otherwise, repository hyperparameters (pertaining to the neural network architecture and its training) were kept at their default values. The SSM powering the architecture was adapted to a diagonal structure for the state transition matrix $A$. For real SSMs, we began with the same default configuration as for complex SSMs, and extended the configuration by performing a grid search over different optimizers (Adam [29], AdamW [36], SGD [45]) and initialization schemes (diagonal-real, diagonal-linear, and legs, as defined in the official S4 repository). We ran three random seeds with the default configuration, and a single random seed with all others.

## F.3  Selectivity

**Copy task.**  Our version of copy task has a delay of $t = 64$. It entails input sequences of length $2t$, where tokens are sampled independently and uniformly as one-hot vectors of dimension 16. The corresponding output sequences are generated by shifting the input sequences $t$ positions to the right, while introducing zero padding. That is, given an input sequence $[a_1, a_2, \ldots, a_{2t}]$, the corresponding output sequence is $[0, 0, \ldots, 0, a_1, a_2, \ldots, a_t]$. Training and evaluation (measurement of test accuracy) in this task are based on randomly generated examples, where the input is sampled from a uniform distribution. In training, a fresh batch of examples is generated for each iteration, and the loss corresponding to an example is the cross-entropy loss averaged over the last $t$ tokens of the output. In evaluation, the zero-one loss is averaged over the last $t$ tokens of the output across one thousand freshly generated examples.

**Induction-head task.**  In the induction-head task [18], the goal is to teach a model to identify and copy a specific subsequence of the input. Figure 1 illustrates this task, which in our case has an induction head size of $h = 128$. The task entails input sequences of length $3h$, *i.e.*, input sequences with $3h$ tokens. The first (less than $h$) tokens in an input sequence are sampled independently and uniformly as one-hot vectors of dimension 16. These tokens are followed by a special *copy token*,

$c = \mathbf{0}$. After the copy token comes a sequence of $h$ tokens, sampled as before and denoted $\mathbf{X}$. The $2h$th token is another copy token $c$, after which the first to penultimate tokens of the sequence $\mathbf{X}$ repeat. The tokens between the first appearance of $\mathbf{X}$ and the second appearance of $c$ are irrelevant for the task. The output sequence corresponding to the above-described input sequence is equal to the input sequence shifted one position to the left (with the last token of $\mathbf{X}$ placed on the right). Training and evaluation (measurement of test accuracy) in this task are based on randomly generated examples, where the following aspects of the input are sampled from uniform distributions: the location of the first copy token $c$, the sequence $\mathbf{X}$, and the irrelevant tokens. In training, a fresh batch of examples is generated for each iteration, and the loss corresponding to an example is the cross-entropy loss averaged over the last $h$ tokens of the output. In evaluation, the zero-one loss is averaged over the last $h$ tokens of the output across one thousand freshly generated examples.

**Hyperparameters.** Our implementation is based on a widely adopted Mamba repository, available at `https://github.com/alxndrTL/mamba.py` (MIT license). Unless stated otherwise, repository hyperparameters (pertaining to the neural network architecture and its training) were kept at their default values. The Mamba neural network was configured to have two layers with a hidden dimension of $64$. We trained the network with a batch size of $8$ and a learning rate of $1 \times 10^{-3}$. Training comprised $1{,}000{,}000$ steps on the copy task, and $250{,}000$ steps on the induction-head task (the latter task warranted a shorter run-time since it was experimented with much more). For complex parameterization, we changed the state transition matrix $A$, the input matrix $B$ and the output matrix $C$ to be complex, while keeping the discretization parameter $\Delta$ real. To maintain parameter count, the dimension of SSMs with complex parameterization was half the dimension with real parameterization, namely, $8$ as opposed to $16$.

