# OpenReview forum: "Provable Benefits of Complex Parameterizations for Structured State Space Models"
_NeurIPS.cc/2024/Conference — NeurIPS 2024 poster_

### Official Review · Reviewer_UNm3 · 2024-06-12

**Soundness:** 3
**Presentation:** 3
**Contribution:** 2
**Rating:** 5
**Confidence:** 4

**Summary:**

This paper provides a theoretical analysis of why SSM needs to be parameterized by complex numbers instead of real numbers. It shows that there exist complex LTI systems that could not be well-approximated by real systems of comparable size. Moreover, it proves that certain dynamics cannot be approximated by real LTI systems unless it has an exponentially large state-space dimension or large system matrices; yet, using complex LTI systems resolves this issue.

**Strengths:**

1. The theoretical statements are precisely made. The sketch of the proofs are helpful for understanding the paper.
2. The comparison between real and complex is fairly thorough, encompassing different perspectives.

**Weaknesses:**

1. My main concern is about the contribution of this work. While it is true that many ML models use real parameterizations, the diagonal matrix $\mathbf{A}$ comes from diagonalizing a general state matrix. Therefore, unless one puts restrictions on the matrix to be diagonalized (e.g. Hermitian), it is natural to assume that $\mathbf{A}$ should be complex-valued. Showing why a real parameterization does not work well sounds like a slightly artificial problem and adds relatively little to the SSM community.
2. The experiments could not strongly corroborate the theory. In addition to showing the performance, maybe some synthetic experiments would be helpful to show the exponential gap in Theorem 2 and Proposition 3.

**Questions:**

1. You formulate the problem using the discrete LTI system. Of course, in SSMs, the parameters are from the continuous-time LTI systems. While this does not change the basic question you are exploring because the real axis in the left half-plane gets mapped to the real axis in the unit disk under virtually all discretization schemes, would Theorem 2 be changed if you take the discretization into account?
2. The paper studies two cases: $\mathbf{A}$, $\mathbf{B}$, and $\mathbf{C}$ are either all restricted to real or all allowed to be complex. Intuitively, however, the important thing is that $\mathbf{A}$ has to be complex. Have you looked into the case where $\mathbf{A}$ is complex and $\mathbf{B}$ and $\mathbf{C}$ are real? In that case, which world would it fall into?
3. In section 3.3, instead of giving two examples of dynamics that are poorly approximated by real systems, maybe there could be a discussion of what you called forward difference. It would be helpful to relate Theorem 2 to some general (and easily interpretable) properties of the dynamics.

**Limitations:**

None.

---

> ### Author Rebuttal · Authors · 2024-08-07
>
> Thank you for your review, and for highlighting the thoroughness and preciseness of our theory. We address your comments and questions below. If you find our responses satisfactory, we would greatly appreciate it if you would consider raising your score.
>
> ## Significance of real diagonal SSMs:
> Real diagonal SSMs (with and without selectivity) underlie some of the most prominent neural network architectures for long sequence processing. This includes the recent Mamba architecture and its variants, as well as other architectures [A, B, C, D, E]. Advantages of real diagonal SSMs include simplicity and speed. The extent to which they compare to complex diagonal SSMs in terms of accuracy has been the subject of much debate (see our introduction). This debate is what motivated our work.
>
> We hope the above addresses your concern regarding the significance of our contribution. If not, please let us know and we will gladly elaborate.
>
> --
>
> **[A]** “Mamba: Linear-time sequence modeling with selective state spaces”, Gu and Dao, 2023
>
> **[B]** “Mega: Moving Average Equipped Gated Attention”, Ma et al., 2023
>
> **[C]** “Transformers are SSMs: Generalized Models and Efficient Algorithms Through Structured State Space Duality”, Dao and Gu, 2024
>
> **[D]** “Megalodon: Efficient LLM Pretraining and Inference with Unlimited Context Length”, Ma et al., 2024
>
> **[E]** “Jamba: A Hybrid Transformer-mamba language model”, Lieber et al., 2024
>
> ## Experimentation:
> Following your comment, we conducted several experiments, including:
> * Demonstration of the gap in practical learnability between real and complex SSMs in the theoretically analyzed setting (i.e., in the setting for which we established such a gap).
> * Demonstration that complex parameterizations for (non-selective) SSMs improve performance of prominent neural network architectures (S4) on real-world data (sequential CIFAR-10, included in the standard long range arena benchmark).
> * Ablation study showing that among the architectural differences between S6 and S4, the one primarily responsible for closing the gap between real and complex parameterizations for SSMs, is the selectivity of the input and output matrices $B$ and $C$.
>
> ***The results of these experiments can be found in the PDF attached to our global response***.
>
> ## Impact of discretization:
> Our theory essentially applies as is (more precisely, it applies given slight modifications in the established bounds) to cases where the SSMs include conventional discretizations. For example, with the discretization laid out in the original Mamba paper [10]: the state transition matrix $A$ is represented as the exponent of a parameter matrix $A’$ times a scalar $\Delta$; and the input matrix $B$ is represented as a parameter matrix $B’$ times $\Delta$ times some matrix with spectral norm no greater than one. All of our proofs readily adapt to this case (in particular, the proofs of Theorem 2, Corollary 1 and Corollary 2 readily adapt to establishing an exponential lower bound on the magnitudes of $( B’ , C )$ for the real SSM).
>
> We will add a discussion of discretization to the camera-ready version. Thank you very much for raising the matter!
>
> ## Complex $A$ with real $B$ and $C$:
> Your intuition is correct. Namely, all benefits we have proven for the complex SSM apply (given a slight modification in one of the established bounds) when the input matrix $B$ and the output matrix $C$ are constrained to be real. To see this, note that our proofs of Propositions 1, Proposition 2 and Theorem 1 do not make use of complex assignments for $B$ and $C$. Our proof of Proposition 3 does make use of such assignments, but can readily avoid them by replacing the discrete Fourier transform with the discrete cosine transform (in this case a factor of $\sqrt{2}$ is introduced in the second bound). We will mention this in the camera-ready version. Thank you for the question!
>
> ## Mappings poorly approximated by the real SSM:
> While Corollary 1 indeed provides an example (albeit a canonical one) of a mapping poorly approximated by the real SSM, Corollary 2 goes far beyond an example, in the sense that it applies to a random (generic) mapping. Nonetheless, we agree with you that it would be interesting to relate the condition brought forth by Theorem 2 — namely, forward differences not being especially small — to easily interpretable properties of the mappings. We invested considerable efforts trying to establish such a relation (primarily using tools from complex analysis, e.g. Rice’s method), but to no avail. We will mention this pursuit as a direction for future work in the camera-ready version. Thank you for raising the matter!

---

> > ### Comment · Reviewer_UNm3 · 2024-08-12
> >
> > Thank you for your response! The additional experiments look convincing; therefore, I have raised my score. My primary concern is still that the underlying question studied in this work is relatively trivial from a linear algebra perspective, pinning that matrices with real eigenvalues are not universal approximators. This prevents me from further increasing my evaluation. However, I acknowledge that the problem studied in this work is important and the author(s) have done a good job of making a point.

---

> > > ### Author Response · Authors · 2024-08-13
> > >
> > > Thank you.
> > >
> > > We respectfully disagree with the following statement:
> > >
> > > > “the underlying question studied in this work is relatively trivial from a linear algebra perspective, pinning that matrices with real eigenvalues are not universal approximators”.
> > >
> > > It is true that matrices with complex eigenvalues are more general than ones with real eigenvalues, but to our knowledge, this does not establish any benefit of complex parameterizations over real parameterizations for diagonal SSMs. Indeed (perhaps counter-intuitively), we show in Proposition 1 that if the dimension of a diagonal SSM is not bounded, real and complex parameterizations are equivalent, in the sense that they both lead to universal approximation (of LTI mappings). The question is then whether complex parameterizations allow approximating mappings with lower dimension or parameter magnitudes than are required with real parameterizations. To our knowledge, this question — which we affirm in Theorems 1 and 2, Corollaries 1 and 2, and Proposition 3 — cannot be answered via simple linear algebraic arguments as you mentioned.
> > >
> > > We hope the above clarification addresses your remaining concern. Please let us know if this is not the case and we will further elaborate.

---

> > > > ### Comment · Reviewer_UNm3 · 2024-08-13
> > > >
> > > > Dear Author(s),
> > > >
> > > > I understand your point regarding real LTI systems with arbitrarily large state dimensions $N$ being universal approximators. I realize that my previous comment about universal approximation should have been more precise, as I was specifically referring to the approximation of matrices with a fixed dimension.
> > > >
> > > > However, this does not alter my primary concern about the relevance of the problem addressed in this paper. From a backpropagation perspective, it's logical to use real parameterization for simplicity, and in that context, this paper serves as a validation that complex numbers are indeed necessary. From a linear algebra standpoint, constraining a matrix to have real eigenvalues is not typically a natural approach, making this paper a (well-executed) exploration of an unusually framed problem. Whether or not real parameterizations are universal approximators does not fundamentally change the nature of the issue at hand.
> > > >
> > > > This is not a critique of the depth of your research. I believe the work presented is of high quality. My comments are more focused on the significance of the problem studied in the paper. I will maintain my current evaluation. Thank you.
> > > >
> > > > Reviewer UNm3

---

> > > > > ### Author Response · Authors · 2024-08-14
> > > > >
> > > > > Thank you for engaging in the discussion and for highlighting the quality of our work! With regards to the motivation behind it: we will include in the camera-ready version a detailed account covering all of our arguments from the discussion (and more).

---

### Official Review · Reviewer_sPCd · 2024-06-13

**Soundness:** 3
**Presentation:** 3
**Contribution:** 3
**Rating:** 8
**Confidence:** 3

**Summary:**

This paper considers learning LTI systems with bounded response. It shows that if we restrict to SSMs with diagonal dynamics, both real-valued and complex valued state sizes suffice. However, it shows that if we use only real-valued SSMs, to learn a sequence of length t we will need parameters that scale as $\exp(t)$, while if we use complex-valued SSMs, we need only parameters that scale as $t$.

**Strengths:**

This paper did a great job of distinguishing between model expressivity (showing that with large enough state size both real- and complex-valued SSMS are equally expressive) and practical learnability (showing that real- valued SSMs need an impractically large state size).

I also liked how their proof of practical learnability was basically based on the point that for stable real-valued diagonal SSMs, the dynamics pretty have to be decaying exponentials, so we will need exponentially many of them to form a basis for a function. On the other hand, for complex-valued diagonal SSMs, we effectively have a Fourier basis, which is more expressive.

I also like the robust limitations section of the paper, which emphasized that although the paper has demonstrated the need for complex parameterization in the LTI setting, they haven't addressed the selectivity setting, leaving open theoretical contributions to explain why Mamba can get away with using only real parameterization for language tasks.

I also liked the proof technique based on impulse series and impulse response, that was a very nice framing.

This paper provides an effort grounded in both experiments and theory that attempts to resolve an important question practitioners genuinely care about.

**Weaknesses:**

The paper would have been more interesting if it had addressed what is going on in the setting of selectivity. On lines 66-7, the authors write "We believe our results may serve as an important stepping stone in this pursuit." It would be nice if more evidence was given. I can't see how the proof techniques carry over. It would be great if some indication could be given of how this paper could be helpful in addressing selectivity.

Footnote 1 is extremely confusing. It basically contradictions lines 101-2, which state that the SSMs considered in the paper are going to be diagonal. So, the eigenvalues of a real-valued diagonal A are definitely always real. I would just remove footnote 1.

**Questions:**

* I think the first sentence of the paper should name check S5 between S4 and Mamba (which is sometimes referred to in the body of its text as S6). S5 helped to pioneer the use of diagonal complex dynamics in an LTI setting, about which this entire paper is based. I think not namechecking it, and instead lopping in onto the end as "and more" does a disservice to a reader trying to learn more about the literature.
* How do the results in this paper serve as an important stepping stone towards understanding real vs complex parameterization in LTV or selective settings?
* What happens if we use a feedthrough matrix D in (1), which is common in nearly all SSM parameterizations?
* What is $\Omega(t)$ on line 187? Is it defined somewhere in the paper?
* Is it possible to do better than $n_{\mathbb{C}} = t$? why or why not?

**Limitations:**

yes

---

> ### Author Rebuttal · Authors · 2024-08-07
>
> Thank you for your support, and for highlighting the importance of the question we study and the merit of our theory and experiments! We address your comments and questions below.
>
> ## Addressing selectivity:
> We agree with you that addressing selectivity — i.e., theoretically supporting the success of real parameterizations in selective SSMs — is an interesting pursuit. Below we qualitatively discuss the reasoning behind our belief that “our results may serve as an important stepping stone in this pursuit”. A more detailed version of this discussion will be added to the paper. Thank you for raising the matter!
>
> Roughly speaking, the separations we established between real and complex SSMs result from a gap in their ability to express oscillations, i.e., to express frequency components in their impulse response: while the complex SSM can easily express any frequency, the real SSM is required to have exponential dimension or parameter magnitudes. Adding selectivity to the real SSM means that its parameters become input-dependent, leading to what can be viewed as an input-dependent impulse response. It can be shown that this dependence allows “importing” frequency components from the input to the impulse response. Therefore, if the input data is sufficiently rich in terms of its frequency content, selectivity may endow real parameterizations with all the benefits we proved for complex parameterizations.
>
> Recall that, as stated in our introduction, [10] conjectured that complex parameterizations are preferable for continuous data modalities (e.g., audio, video), whereas for discrete data modalities (e.g., text, DNA) real parameterizations suffice. The explanation above aligns with this conjecture: continuous data modalities typically consist of low frequencies only, whereas discrete data modalities have a “whiter spectrum,” i.e. a more uniform mix of frequencies.
>
> As stated, a detailed version of the above discussion will be added to the paper.
>
> ## Footnote 1:
> Per your suggestion, we will remove Footnote 1.
>
> ## Questions (addressed by order of appearance):
> * We will explicitly mention S5 when listing prominent SSM-based neural network architectures.
> * See “addressing selectivity” section above.
> * Our theory essentially applies as is (more precisely, it applies given very slight modifications in the established bounds) to the case where SSMs include a feedthrough $D$. The reason why incorporation of $D$ does not make a material difference is that it can easily be emulated by: (i) increasing the dimension of the SSM by one; (ii) assigning one to the last diagonal entry of $A$ and the last entry of $B$; and (iii) assigning $D$ to the last entry of $C$. We will mention this in the camera-ready version. Thank you for raising the matter!
> * $\Omega (\cdot )$ in line 187 stands for Big-Omega notation, i.e. it signifies a function that is at least linear in its argument. Note that it is used to qualitatively characterize a dependence whose precise form is provided by Theorem 2.
> * In Proposition 3 (and Proposition 1), it is indeed possible to improve (relax) the requirement $n_{\mathbb{C}} \geq t$. Namely, by leveraging the fact that the discrete Fourier transform of a real sequence is symmetric, one can update the proof of Proposition 3 to show that $n_{\mathbb{C}} \geq \lceil t / 2 \rceil$ suffices. Parameter counting arguments imply that the latter requirement is tight. We will mention all of this in the camera-ready version. Thank you for asking about it!

---

> ### Comment · Reviewer_sPCd · 2024-08-07
> **Increasing score to 8**
>
> Thank you for a very clear and interesting rebuttal.
>
> I think your discussion of selectivity is very interesting and should be included in as much details as possible in the camera ready version.
>
> Your new experiments in the .pdf are great as well! They should also be included in camera ready.
>
> I am raising my score to an 8.

---

> > ### Author Response · Authors · 2024-08-09
> >
> > Thank you very much! The camera ready version will indeed expand on both selectivity and the new experiments.

---

### Official Review · Reviewer_SDv4 · 2024-07-07

**Soundness:** 4
**Presentation:** 4
**Contribution:** 3
**Rating:** 7
**Confidence:** 3

**Summary:**

This work establishes a formal gap between real and complex parameterizations of stable, diagonal SSMs. While complex parameters can trivially express any real SSM, the converse is not true and real SSMs need an arbitrarily large number of parameters to approximate complex SSMs in at least two important cases.

**Strengths:**

1. Clear writing and presentation
2. Theoretical results support a clear practical suggestion: use complex parametrizations for your SSM if you don't have input selectivity.

**Weaknesses:**

1. Experimental results on non-synthetic datasets would be great. Especially considering the theorems regarding practical learnability and the exponentiality of real parametrizations for random impulse responses with high probability.

**Questions:**

See above.

**Limitations:**

The paper has a thorough discussion of the limitations of the analysis.

---

> ### Author Rebuttal · Authors · 2024-08-07
>
> Thank you for your support!
>
> We conducted several additional experiments, including:
> * Demonstration of the gap in practical learnability between real and complex SSMs in the theoretically analyzed setting (i.e., in the setting for which we established such a gap).
> * Demonstration that complex parameterizations for (non-selective) SSMs improve performance of prominent neural network architectures (S4) on real-world data (sequential CIFAR-10, included in the standard long range arena benchmark).
> * Ablation study showing that among the architectural differences between S6 and S4, the one primarily responsible for closing the gap between real and complex parameterizations for SSMs, is the selectivity of the input and output matrices $B$ and $C$.
>
> ***The results of these experiments can be found in the PDF attached to our global response***.

---

> > ### Comment · Reviewer_SDv4 · 2024-08-07
> >
> > Thank you for the response and additional results. I will stay with my assessment.

---

> > > ### Author Response · Authors · 2024-08-09
> > >
> > > Thank you.  Please let us know if a need for further information arises.

---

### Official Review · Reviewer_9K2D · 2024-07-11

**Soundness:** 4
**Presentation:** 3
**Contribution:** 3
**Rating:** 7
**Confidence:** 5

**Summary:**

The paper deals with an important question: is it possible to show a concrete advantage of complex diagonal RNNs compared to real diagonal RNNs? The motivation is clear, especially for people who studied the SSM literature. While modern SSM variants used in language modeling (e.g. S6-Mamba) do not make use of complex numbers, for some reasoning tasks in the long-range arena, these are necessary. This motivates the question "which tasks necessitate complex numbers and why?". The authors start with a useful recap, and correctly point out universality of both real and complex linear RNNs in the approximation of filters. Then show one example of a filter which is represented easily with complex numbers, but requires a huge hidden state if the recurrence is real. Further, the authors discuss a bigger class of filters that comprises random convolutions and shifts: here, real RNN parameters have to explode to reach the correct values. The authors corroborate their findings with experimental validations.

**Strengths:**

I believe in the value of complex numbers in recurrent networks.  I liked reading the paper, and I think many people can find it interesting. However, the theoretical results might not be super surprising for readers who already worked on theory of RNNs (see next part) but experiments are interesting. I think it's good to draw attention to the question the authors study.
Pros: Math is at a good level of formality, notation is precise, and the appendix is well structured. Clarity is good, though a specific thing can be improved (see later). I think with some tweaks, this can be a very good paper. I especially liked some parts of Sec. 3.3.

**Weaknesses:**

I like the theme of this paper and appreciate the formality and correctness of the results. However, I think some improvements are possible and some things need to be clarified.

- Theorem 1: this is a crucial piece of Section 3. In short, both real and complex diagonal RNNs are universal, but complex diagonal linear RNNs require exponentially fewer parameters to approximate some specific functions. I agree with the proof, which is very simple: you cannot approximate well sin(x) using exponential modes (i.e. real RNNs). It is good to remind this fact, though the result is not novel it helps the discussion. However, while your statement is correct, it might be misleading: it looks like for a big part of the functions of interest in the hypothesis class, complex numbers are crucial. What the theorem actually shows is that for a subset of measure zero in the class of smooth functions, complex valued recurrences are more effective. I agree oscillations are important, but this is not formal and should be toned down. I would phrase this as an example rather than a theorem. It is indeed just a counterexample.

- The second part of Section 3, Sec 3.3, is completely disconnected: here, you are talking about parameter magnitudes. This is fine, but I was a bit confused by the wall of text at the beginning of page 5. You should be much more schematic. You are now talking about something bit different, and the bridge should be clear but concise.

- Sec 3.3 is interesting, but I'm not at all convinced that the results you found are orthogonal to those of Orvieto et al. (discussed at the end of page 8). I think your results are stronger, but comparing further makes your paper much more robust: Orvieto et al. shows that to retrieve information from real RNNs hidden states, the readout maps must explode in magnitude. Here, you found something similar, but on the B & C matrices when approximating some filters. I suspect the mechanism is similar, do you agree? I think drawing a better connection is needed, no way those effects have a disjoint cause.

- Sec 3.3: All bounds you have here should be validated in the simplest toy setting possible. Would it be possible to verify empirically all your bounds in the setting of the propositions? This makes the reader believe much more in the results and makes the bounds clearer and visual.

- Experiments: I like them, but as the authors themselves admit, more efforts are needed. Well my question here is: what would you do next, and why do you think the paper should be accepted without further experiments?

**Questions:**

See above, + I have another question - something which I might have missed but is my biggest curiosity:

- How do you explain the fact that Mamba on language modeling works better with real recurrences? I am perfectly aware the question is not easy - but given your efforts, I believe your paper should try to answer this question. What you claim in the paper is "selectivity allows the real parameterization to achieve comparable performance": this is a bit too quick, on an extremely crucial issue. I expected the discussion to be much more precise on this point.

I will be active in the rebuttal, and will engage in the discussion so to update my borderline score. I thank the authors in advance for their efforts. I know the topic in this paper is not an easy one, but for this exact reason I care the findings are robust and discussion complete.

**Limitations:**

See above

---

> ### Author Rebuttal · Authors · 2024-08-07
>
> Thank you for the thoughtful review, for highlighting that “with some tweaks, this can be a very good paper”, and for the willingness to “engage in the discussion so [as] to update [your] borderline score”. Below we address your comments and questions.
>
> ## Specificity of Theorem 1:
> Theorem 1 readily extends to any function realized by the complex SSM whose impulse response oscillates. More precisely, the proof of Theorem 1 can easily be extended to establish a lower bound on $n_{\mathbb{R}}$ that is linear in $t$ whenever the restriction to odd elements of the impulse response of the complex SSM admits a number of sign changes linear in $t$. With this extension, Theorem 1 applies to a positive measure subset of the complex SSM’s function class. Moreover, among the subset of the complex SSM’s function class which is characterized by $A_{\mathbb{C}}$ having entries close to the unit circle, the above-described extension of Theorem 1 can be shown to apply to “most” functions. We will elaborate on this in the camera-ready version. Thank you for bringing it up!
>
> Notwithstanding the above, we note that proving separation in expressiveness through particular counterexamples is very common. Indeed, various important results in deep learning theory fall under this category (see references in “On the Expressive Power of Deep Learning: A Tensor Analysis” by Cohen et al.).
>
> ## Opening of Section 3.3:
> Following your comment, this text was completely refactored. In particular, we broke down Section 3.3 to three subsubsections, titled: “Real Parameterizations Suffer from Exponentiality”, “Exponentiality Impedes Practical Learning”, and “Complex Parameterizations Do Not Suffer from Exponentiality”. We believe Section 3.3 is much clearer now. Thank you for the feedback!
>
> ## Relation to Orvieto et al.:
> Thank you for raising this point! Upon closer inspection, we indeed believe that the result of Orvieto et al. ([25], Section 4.1) can be viewed as a special case of ours. In particular, the task considered in Orvieto et al. — namely, reconstructing past inputs to a real SSM from its current state using a linear mapping — can be viewed as taking an output matrix $C_{\mathbb{R}}$ with which the input-to-output mapping of a real SSM is a canonical copy (delay) mapping. Our Corollary 1 implies that this necessitates exponential parameter magnitudes, therefore essentially recovers the result of Orvieto et al. We stress that our Theorem 2 is far more general than our Corollary 1 (and accordingly, than the result of Orvieto et al.). Indeed, our Theorem 2 applies (i.e., ensures exponential parameter magnitudes for the real SSM) not only with a copy input-to-output mapping, but with any input-to-output mapping whose impulse response has forward differences that are not especially small (this includes, e.g., random input-to-output mappings — see Corollary 2).
>
> We will detail the relation between the result of Orvieto et al. and ours in the camera-ready version. Thanks again!
>
> ## Empirical demonstration of bounds in Section 3.3:
> Following your request, we conducted experiments demonstrating the bounds in Section 3.3. ***The results of these experiments can be found in the PDF attached to our global response***. In a nutshell, the results show that while the complex SSM is able to learn all impulse responses (up to times no greater than its dimension), when the impulse response is such that the real SSM is provably required to have exponential parameter magnitudes, training the real SSM does not converge.
>
> ## Further experiments:
> Following your feedback, we conducted further experiments, including:
> * Demonstration that complex parameterizations for (non-selective) SSMs improve performance of prominent neural network architectures (S4) on real-world data (sequential CIFAR-10, included in the standard long range arena benchmark).
> * Ablation study showing that among the architectural differences between S6 and S4, the one primarily responsible for closing the gap between real and complex parameterizations for SSMs, is the selectivity of the input and output matrices $B$ and $C$.
>
> ***The results of these experiments can be found in the PDF attached to our global response***.
>
> ## Success of real parameterizations in selective SSMs:
> We agree with you that theoretical support for the success of real parameterizations in selective SSMs is an extremely crucial pursuit. As stated in the paper, we believe our results may serve as an important stepping stone in this pursuit. Below we qualitatively discuss the reasoning behind our belief. A more detailed version of this discussion will be added to the paper. Thank you for raising the matter!
>
> Roughly speaking, the separations we established between real and complex SSMs result from a gap in their ability to express oscillations, i.e., to express frequency components in their impulse response: while the complex SSM can easily express any frequency, the real SSM is required to have exponential dimension or parameter magnitudes. Adding selectivity to the real SSM means that its parameters become input-dependent, leading to what can be viewed as an input-dependent impulse response. It can be shown that this dependence allows “importing” frequency components from the input to the impulse response. Therefore, if the input data is sufficiently rich in terms of its frequency content, selectivity may endow real parameterizations with all the benefits we proved for complex parameterizations.
>
> Recall that, as stated in our introduction, [10] conjectured that complex parameterizations are preferable for continuous data modalities (e.g., audio, video), whereas for discrete data modalities (e.g., text, DNA) real parameterizations suffice. The explanation above aligns with this conjecture: continuous data modalities typically consist of low frequencies only, whereas discrete data modalities have a “whiter spectrum,” i.e. a more uniform mix of frequencies.

---

> > ### Comment · Reviewer_9K2D · 2024-08-09
> > **Thanks!**
> >
> > Dear Authors,
> > Thanks so much for the efforts in this rebuttal. I raised my score to accept as I believe the additional experiments, along with your interpretation of selectivity, are convincing.
> >
> > I still would tone down Thm 1 and place it as a counterexample. What you can do is argue that oscillations are important, but you cannot be 100% formal in this. A thing that would be super nice to show is that using only exponential decay (real numbers) leads to a basis that requires, in general, more elements compared to exp + exp*sine waves. This may be true, but it requires some spectral or functional analysis. It would make your point, though, SO MUCH stronger. Currently, Thm1 gives a hint, but is not giving us the generality expected from a theorem.

---

> > > ### Author Response · Authors · 2024-08-10
> > >
> > > Thank you very much!
> > >
> > > As you suggest, we will reposition Theorem 1 and clarify its limitations.
> > >
> > > With regards to the stronger result you outline, there are two steps towards it which we intend to take:
> > > * Extending the result in Theorem 1 as discussed in our rebuttal. Namely, extending it to apply to sufficiently oscillating impulse responses, which form a positive measure subset of the complex SSM’s function class.
> > > * Analyzing forward differences of oscillatory impulse responses, thereby hopefully drawing another corollary of Theorem 2, which will establish that in order to approximate an oscillatory function, the real SSM must have dimension or parameter magnitudes exponential in $t$.
> > >
> > > Thank you again for your thorough analysis and very useful suggestions!

---

### Author Rebuttal · Authors · 2024-08-07

We thank all reviewers for their time and feedback, addressed per reviewer in our individual responses.

***Attached to this comment is a PDF presenting results of new experiments*** which will be added to the paper. These experiments include:
* Demonstration of the gap in practical learnability between real and complex SSMs in the theoretically analyzed setting (i.e., in the setting for which we established such a gap).
* Demonstration that complex parameterizations for (non-selective) SSMs improve performance of prominent neural network architectures (S4) on real-world data (sequential CIFAR-10, included in the standard long range arena benchmark).
* Ablation study showing that among the architectural differences between S6 and S4, the one primarily responsible for closing the gap between real and complex parameterizations for SSMs, is the selectivity of the input and output matrices $B$ and $C$.

---

### Decision · Program_Chairs · 2024-09-25

**Decision:**

Accept (poster)

**Comment:**

This paper investigates whether complex diagonal State Space Models (SSMs) have an advantage over real diagonal SSMs. It demonstrates that complex parameterization in SSMs is important because certain dynamics cannot be well-approximated by real-valued models without requiring a large state space or large system matrices. Furthermore, the paper shows that complex SSMs need significantly fewer parameters to learn specific systems compared to real SSMs, particularly for long sequences.

As reviewer UNm3 noted, the question being addressed may seem somewhat unnatural at first, since diagonalization of a general state matrix already results in a diagonal matrix with complex entries. It seems intuitive that restricting the system to real values would hurt performance --- not only due to reduced parameter count but also because the dynamics that can be modeled by a real-valued diagonal matrix are quite limited. Complex values are essential for capturing oscillatory behavior, and reducing these values toward zero implies heavy damping of the system. Thus, the paper's findings are rather intuitive and may not surprise those familiar with dynamical systems theory. Similarly, reviewer 9K2D points out that "the theoretical results might not be super surprising for readers who already worked on the theory of RNNs." Indeed, similar questions have been widely studied in the past.

Nevertheless, the authors make a valid point that there is an ongoing debate, likely driven by engineering perspectives, on the practical advantages of complex versus real parameterizations in SSMs. In this regard, the paper contributes insights that could help steer this discussion.

Overall, the reviewers are positive, acknowledging both the strengths and weaknesses of the work. While I agree that the paper provides interesting insights, I find the findings and experiments somewhat limited, and I do not believe the paper will have a significant impact in its current form. However, it could be much stronger if the authors address the reviewers' comments in the camera-ready version. Nevertheless, I suggest accepting this paper.